# Global curvature for second-order optimization of neural networks

**Alberto Bernacchia** [1]

## Abstract

Second-order optimization methods, which leverage the local curvature of the loss function, have the potential to dramatically accelerate the training of machine learning models. However, these methods are often hindered by the computational burden of constructing and inverting large curvature matrices with $\mathcal{O}(p^2)$ elements, where $p$ is the number of parameters. In this work, we present a theory that predicts the *exact* structure of the global curvature by leveraging the intrinsic symmetries of neural networks, such as invariance under parameter permutations. For Multi-Layer Perceptrons (MLPs), our approach reveals that the global curvature can be expressed in terms of $\mathcal{O}(d^2 + L^2)$ independent factors, where $d$ is the number of input/output dimensions and $L$ is the number of layers, significantly reducing the computational burden compared to the $\mathcal{O}(p^2)$ elements of the full matrix. These factors can be estimated efficiently, enabling precise curvature computations. To evaluate the practical implications of our framework, we apply second-order optimization to synthetic data, achieving markedly faster convergence compared to traditional optimization methods. Our findings pave the way for a better understanding of the loss landscape of neural networks, and for designing more efficient training methodologies in deep learning. Code: github.com/mtkresearch/symo_notebooks

## 1. Introduction

Neural network models are commonly trained using adaptive variants of gradient descent and momentum (Schmidt et al., 2021). Second-order optimization methods, which exploit the curvature of the loss function, have shown the potential for significantly faster convergence compared to

first-order approaches (Bottou et al., 2018). These methods face a major challenge: the need to compute and invert curvature matrices of size $p \times p$, where $p$ is the number of model parameters. Several studies have explored block-diagonal and Kronecker-factored approximations for curvature computation in neural networks (Martens & Grosse, 2015; Eschenhagen et al., 2024). Recent advances have scaled these methods effectively (Ba et al., 2017; Anil et al., 2021; Kasimbeg et al., 2025), even enabling second-order pre-training of large language models (Liu et al., 2025). However, despite their computational efficiency, these approximations lack strong theoretical guarantees for nonlinear problems (Bernacchia et al., 2018; Karakida & Osawa, 2020). Exact second-order optimization, avoiding block-diagonal approximations, remains tractable only for small-scale models (Cai et al., 2019; Arbel et al., 2023; Korbit et al., 2024) or specific architectures, e.g. reversible neural networks (Buffelli et al., 2024).

Most neural network architectures are built on a layered structure, where the core operations consist of matrix-vector products combined with relatively simple nonlinearities. Due to this structural framework, neural networks exhibit well-known symmetries: for example, permuting the rows of a parameter matrix while applying the same permutation to the columns of the matrix in the subsequent layer leaves the overall computation of the network unchanged, as illustrated in Figure 1 (Hecht-Nielsen, 1990; Chen et al., 1993). While all neural networks exhibit symmetries, the specific symmetries vary by architecture. This raises a natural question: can we harness the unique symmetries of a given neural network architecture to design a tailored optimization algorithm? Our key contribution is demonstrating that, by leveraging the precise symmetries of a model, we can compute the cur-

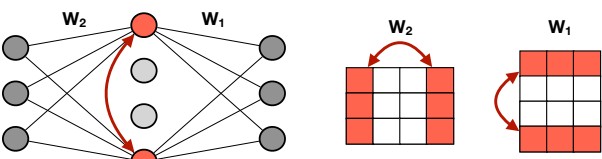

*Figure 1.* **Illustration of permutation symmetry**. The output of a neural network is invariant for swapping any pair of neurons within a layer. That corresponds to simultaneously swapping the rows of incoming and the columns of outgoing weight matrices.

---

[1]MediaTek Research, Cambridge, UK. Correspondence to: Alberto Bernacchia <alberto.bernacchia@mtkresearch.com>.

*Proceedings of the 42nd International Conference on Machine Learning*, Vancouver, Canada. PMLR 267, 2025. Copyright 2025 by the author(s).

vature at significantly reduced computational cost, making second-order optimization computationally feasible.

While prior work has leveraged continuous symmetries to enhance optimization (Neyshabur et al., 2015; Meng et al., 2018; Zhao et al., 2022; 2023b) and to impose constraints on the curvature (Kunin et al., 2020), our study is the first to utilize discrete symmetries for computing the full curvature matrix. A key innovation of our approach is the computation of the *global* curvature, averaged over the parameter space, rather than the *local* curvature, which depends on specific parameter values. The primary motivation for considering global curvature is its efficient computation, which serves as the main contribution of this work. Future research will explore whether global curvature serves as a reliable approximation of local curvature and whether it offers inherent advantages (Yang et al., 2021; Yao et al., 2021; Titsias, 2024). A detailed discussion of the related work is given in Appendix A.

We contribute the following:

- For Multi-Layer Perceptrons (MLPs), we derive an exact expression for the global curvature matrix, revealing that it depends on $\mathcal{O}(d^2 + L^2)$ unknown factors, where $d$ is the number of input/output dimensions and $L$ is the number of layers. This significantly reduces the computational complexity compared to the original matrix size of $\mathcal{O}(p^2)$.

- We analyze the impact of the activation function on global curvature and find that less symmetric functions, such as ReLU, exhibit a more intricate curvature structure compared to more symmetric functions, such as linear or Tanh activations. This increased complexity corresponds to a larger number of unknown factors in the curvature matrix.

- We propose a simple algorithm for estimating the unknown factors of the curvature and for computing the second-order update, that significantly reduces complexity by using a surrogate matrix with $\mathcal{O}(d^2 + L^2)$ elements instead of the $\mathcal{O}(p^2)$ elements of the full matrix.

- We demonstrate the effectiveness of our approach by running second-order optimization on a two-layer MLP and synthetic data. Empirical results show that the second-order update substantially accelerate convergence.

## 2. Background

In this section, we recall known facts that are instrumental for stating our main results, and we also provide a new theorem that may be of independent interest, Theorem 2.2.

In Section 2.1, we review second-order optimization. In Section 2.2 we describe all known symmetries of neural networks, and we motivate our choice for this study. In Section 2.3 we prove that probability distributions remain invariant upon equivariant maps, under a more general case than previously known. In Section 2.4, we describe how the curvature is constrained by the symmetries of the neural network.

### 2.1. Second-order optimization

We consider a scalar loss function $\mathcal{L}(\boldsymbol{\theta})$ of parameters $\boldsymbol{\theta} \in \mathbb{R}^p$. In machine learning, the loss also depends on either a dataset or a data distribution, but we do not make this dependence explicit here. Second order optimization corresponds to the following update

$$\boldsymbol{\theta}_{t+1} = \boldsymbol{\theta}_t - \alpha M_t \nabla \mathcal{L}(\boldsymbol{\theta}_t) \tag{1}$$

where $\alpha$ is the learning rate and $M_t$ is the pre-conditioning matrix, usually the inverse of the curvature matrix. Various studies have employed different formulations for the curvature matrix, including the Fisher information matrix (Martens & Grosse, 2015; Bernacchia et al., 2018; Garcia et al., 2023), the Gauss-Newton matrix (Botev et al., 2017; Yu et al., 2024; Buffelli et al., 2024), the Hessian matrix (Goldfarb et al., 2020), and the gradient covariance matrix (Duchi et al., 2011). Within our framework, all these matrices transform identically under the symmetries of a neural network (see Section 3). While their numerical values differ, they share the same underlying structure, making our approach applicable to any of them.

In most previous work, the pre-conditioning matrix is *local*, it depends on the current value of the parameters, $M(\boldsymbol{\theta}_t)$. Here we take a different approach, similar to Titsias (2024), and we consider global averages of the curvature. In particular, we consider a probability distribution $p_t$ of parameters $\boldsymbol{\theta}$ at training step $t$, which is induced by a distribution of initial conditions on parameters evolving under the same training dynamics. We define the gradient mean and covariance

$$\boldsymbol{\mu}_t = \mathbb{E}_{\boldsymbol{\theta}_t} \nabla \mathcal{L}(\boldsymbol{\theta}_t) \tag{2}$$

$$\Sigma_t = \mathbb{E}_{\boldsymbol{\theta}_t} \nabla \mathcal{L}(\boldsymbol{\theta}_t) \nabla \mathcal{L}(\boldsymbol{\theta}_t)^T - \boldsymbol{\mu}_t \boldsymbol{\mu}_t^T \tag{3}$$

and we set the preconditioning matrix equal to the inverse square root of the gradient covariance

$$M_t = \Sigma_t^{-\frac{1}{2}} \tag{4}$$

The choice of taking the square root is motivated by other adaptive methods, such as Adam (Kingma & Ba, 2014), Adagrad (Duchi et al., 2011), Shampoo (Gupta et al., 2018) and RMSProp (Tieleman & Hinton, 2012). A debate on whether the curvature matrix should be square rooted is

currently ongoing (Lin et al., 2024; Choudhury et al., 2024; Morwani et al., 2025), our experience is that the square root significantly stabilizes training. Appendix B argues that the square root is also a reasonable choice for the toy case of a quadratic loss function.

We note that previous methods take an average of gradients over training iterations or data points, while we define the covariance by an average over an ensemble of models with different initialization. In practice, we estimate the covariance within a single model (see Section 3), we assume that the structure present in the global curvature is also present in the local curvature and captures meaningful variations for optimization purposes.

**2.2. Symmetries of neural networks**

Neural networks exhibit a rich variety of symmetries. The most general and ubiquitous is permutation symmetry (see Figure 1), whereby permuting the rows of a weight matrix and simultaneously permuting the columns of the subsequent layer's weight matrix leaves the network's output unchanged (Hecht-Nielsen, 1990). Permutation symmetry applies to nearly all neural networks and is so fundamental that new architectures often require deliberate design choices to break it (Lim et al., 2024). Beyond permutation symmetry, additional symmetries arise in specific architectures: sign flip for networks with odd activation functions (Chen et al., 1993), rescaling in ReLU networks, exploiting their homogeneity (Neyshabur et al., 2015), translation in Softmax layers (Kunin et al., 2020), scaling in normalization layers (Kunin et al., 2020), general linear transformations of keys and queries in transformers (Ziyin, 2024), deep linear networks (Zhao et al., 2022) and data-dependent transformations (Zhao et al., 2023a). It is unlikely that any other symmetries exist, at least in common neural network architectures (Grigsby et al., 2023; Chen et al., 1993).

In Section 2.3, we introduce a critical assumption underpinning our work. Specifically, we require not only that the neural network output be invariant under a given symmetry group acting on the parameters, but also that the probability distribution over the initial parameters exhibits the same invariance. As reviewed in Appendix C, this condition further necessitates that the transformation must be similar to an orthogonal transformation (Flytzanis, 1977). This additional assumption excludes many of the symmetries discussed earlier, such as scaling, translations and general linear transformations, but retains permutations, sign flips, rotations and reflections. Consequently, we restrict our analysis to these key symmetry groups, with a particular focus on orthogonal transformations.

A Multi-Layer Perceptron (MLP) of $L$ layers is defined by the following expression

$$\mathbf{h}_\ell = W_\ell \, \sigma_\ell(\mathbf{h}_{\ell-1}) + \mathbf{b}_\ell \qquad \ell = 1, \dots, L \qquad (5)$$

where $\sigma_\ell$ is the actibvation function, a pointwise nonlinearity ($\sigma_1$ is the identity), $h_0 = x \in \mathbb{R}^{d_0}$ is the input, $h_L = y \in \mathbb{R}^{d_L}$ is the output and $h_\ell \in \mathbb{R}^{d_\ell}$ is the latent representation of layer $\ell$. The parameters of the neural network are the biases $\mathbf{b}_\ell \in \mathbb{R}^{d_\ell}$ and weights $W_\ell \in \mathbb{R}^{d_\ell \times d_{\ell-1}}$. We consider the following transformation, applying to all layers $\ell = 1, \dots, L$.

$$\mathbf{b}_\ell \longrightarrow V_\ell \mathbf{b}_\ell \qquad (6)$$

$$W_\ell \longrightarrow V_\ell W_\ell V_{\ell-1}^T \qquad (7)$$

where the matrix $V_\ell \in \mathbb{R}^{d_\ell \times d_\ell}$ is assumed orthogonal for each layer, $V_\ell^T = V_\ell^{-1}$. We denote by $G$ the corresponding transformation acting on the set of all parameters, which combines the effect of $V_\ell$ for all layers. This transformation leaves the output of the neural network invariant if it belongs to a symmetry group that depends on the activation function. We consider three cases:

- **Linear activations**. The network is invariant if $V_\ell$ belongs to the set of orthogonal matrices $\mathrm{O}(d_\ell)$, that corresponds to the orthogonal group (rotations and reflections). Across all layers, the transformation $G$ belongs to the product $\mathbb{G}_o = \mathrm{O}(d_1) \times \dots \times \mathrm{O}(d_{L-1})$. The group is continuous and compact, of dimension $\sum_{\ell=1}^{L-1} \frac{d_\ell(d_\ell-1)}{2}$ (Zhao et al., 2022).

- **Odd activations** (e.g. Tanh) The network is invariant if $V_\ell$ belongs to the set of signed permutation matrices $\mathrm{B}(d_\ell)$, that corresponds to the signed symmetric group. Across all layers, the transformation $G$ belongs to the product $\mathbb{G}_b = \mathrm{B}(d_1) \times \dots \times \mathrm{B}(d_{L-1})$. The group is discrete and includes $\prod_{\ell=1}^{L-1} 2^{d_\ell} d_\ell!$ elements (Chen et al., 1993).

- **Other activations** (e.g. ReLU) The network is invariant if $V_\ell$ belongs to the set of permutation matrices $\mathrm{S}(d_\ell)$, that corresponds to the symmetric group. Across all layers, the transformation $G$ belongs to the product $\mathbb{G}_s = \mathrm{S}(d_1) \times \dots \times \mathrm{S}(d_{L-1})$. The group is discrete and includes $\prod_{\ell=1}^{L-1} d_\ell!$ elements (Hecht-Nielsen, 1990).

The three groups satisfy a decreasing set of constraints, $\mathbb{G}_o \supset \mathbb{G}_b \supset \mathbb{G}_s$. Correspondingly, we show in Section 3 that the structure of the curvature increases in complexity as the group gets smaller. We note that there is no symmetry on the input and output side of the MLP, therefore $V_0 = \mathrm{I}_{d_0}$ and $V_L = \mathrm{I}_{d_L}$ are fixed to identity matrices. Additional constraints may be introduced if the data has symmetries, but we do not consider that case in this work.

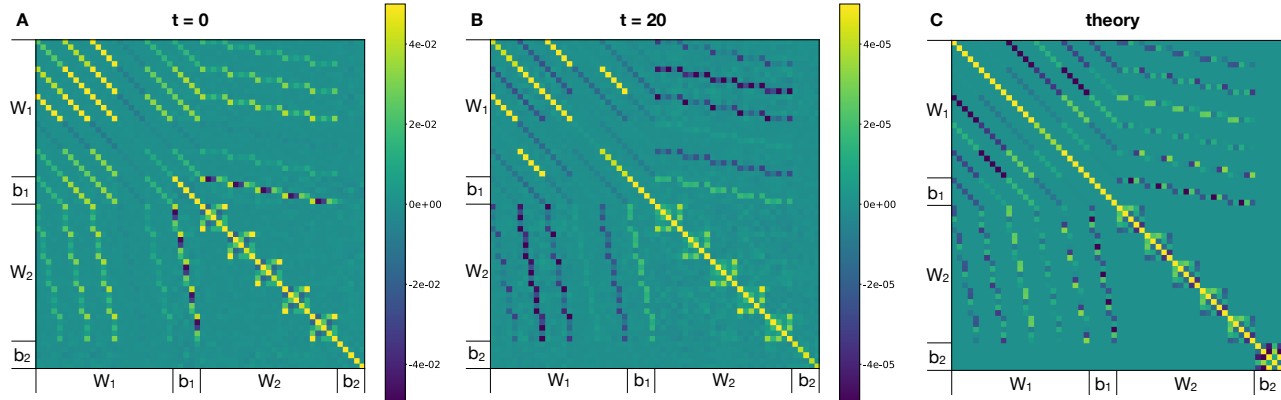

*Figure 2.* **Gradient covariance matrix for two-layer MLP with Tanh activations**, in a tiny model with 5 neurons in each layer. $W_1, W_2$ are the weights and $\mathbf{b}_1, \mathbf{b}_2$ are the biases of the two layers. **A** and **B**: Covariance is measured by averaging 10000 models with different initialization, before training (**A**: $t = 0$) and after 20 steps of gradient descent (**B**: $t = 20$). **C**: Covariance generated by our theory for a random draw of the factors. We highlight that theory predicts the overall structure, rather than the specific numerical values.

Throughout this work, we assume that the loss function depends on the parameters only through the output of the neural network, therefore the loss itself is also invariant for the group action

$$\mathcal{L}(G\boldsymbol{\theta}) = \mathcal{L}(\boldsymbol{\theta}) \quad \forall G \in \mathbb{G} \tag{8}$$

where $\mathbb{G}$ is equal to either $\mathbb{G}_o$, $\mathbb{G}_b$ or $\mathbb{G}_s$.

### 2.3. Invariant distribution throughout training

In this section, we assume that the parameter distribution is invariant under the specified symmetry group at initialization. We then demonstrate that this invariance is preserved throughout training, if the parameter update rule is equivariant with respect to the same symmetry group. Previous studies have established that such invariance holds under the assumption of a globally invertible map (Köhler et al., 2019). Here we extend this result by proving that the same invariance is maintained under the less restrictive assumption of local invertibility, as stated in Theorem 2.2. This generalization broadens the applicability of the result and may be of independent interest.

Parameters are initialized at $t = 0$ according to a probability distribution $p_0(\boldsymbol{\theta}_0)$, and their evolution at subsequent training steps is governed by the update rule

$$\boldsymbol{\theta}_t = \mathbf{u}_t(\boldsymbol{\theta}_{t-1}) \tag{9}$$

At this stage, we do not impose a specific form on the update rule, as the results in this section are derived under general assumptions. However, these results will be utilized for the update rule of Equation (1).

**Assumption 2.1.** The probability distribution on the initial parameters $\boldsymbol{\theta}_0$ is invariant under the action of group $\mathbb{G}$

$$p_0(G\boldsymbol{\theta}_0) = p_0(\boldsymbol{\theta}_0) \quad \forall G \in \mathbb{G} \tag{10}$$

We consider different groups depending on the activation function, as described in Section 2.2, namely $\mathbb{G} = \mathbb{G}_o$, $\mathbb{G}_b$ or $\mathbb{G}_s$. Appendix D shows that Assumption 2.1 is satisfied by the most common initialization routines used in deep learning. In Pytorch for example, Assumption 2.1 holds for `nn.init.normal` and `nn.init.orthogonal`, with all groups $\mathbb{G}_o, \mathbb{G}_b, \mathbb{G}_s$. Also, it holds when using `nn.init.uniform` and `nn.init.sparse` with $\mathbb{G}_b, \mathbb{G}_s$, but not with $\mathbb{G}_o$, and it still holds when using layer-dependent parameters, for example `nn.init.kaiming` and `nn.init.xavier`. See Appendix D for details.

**Theorem 2.2.** *Assume that the update rule $\boldsymbol{\theta}_t = \mathbf{u}_t(\boldsymbol{\theta}_{t-1})$ is differentiable and its Jacobian is non-singular almost everywhere. Furthermore, it is equivariant under a volume-preserving transformation $G$, namely*

$$\mathbf{u}_t(G\boldsymbol{\theta}) = G\mathbf{u}_t(\boldsymbol{\theta}). \tag{11}$$

*Then, if the probability distribution of parameters is invariant at step $t - 1$, then it must be invariant also at step $t$*

$$p_t(G\boldsymbol{\theta}_t) = p_t(\boldsymbol{\theta}_t) \tag{12}$$

The proof is provided in Appendix E. A similar result can be found in Theorem 1 of Köhler et al. (2019), however they assume that the mapping $\mathbf{u}_t$ is globally invertible, which is quite restrictive. Theorem 2.2 considers the more general case in which the mapping is just locally invertible.

The assumption of equivariance of $\mathbf{u}_t$ is satisfied by many optimizers in deep learning, provided that the loss is invariant. Appendix F shows that both gradient descent (with or without momentum) and the second-order update of Equation (1) are equivariant for all groups $\mathbb{G}_o, \mathbb{G}_b, \mathbb{G}_s$, while the Adam optimizer is equivariant for $\mathbb{G}_b, \mathbb{G}_s$, but not $\mathbb{G}_o$.

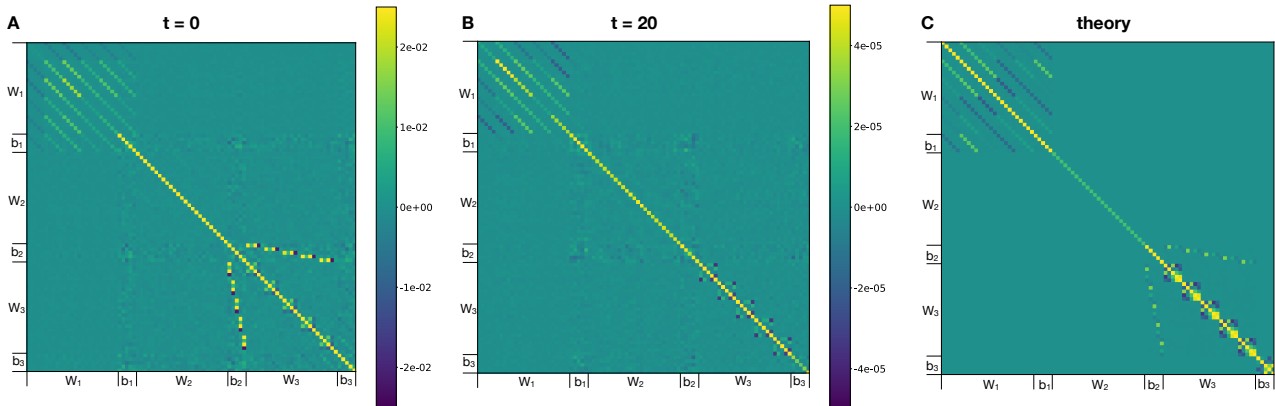

*Figure 3.* **Gradient covariance matrix for three-layer MLP with Tanh activations**, similar to Figure 2 but with three layers. Most of the correlations across layers are lost when adding the third layer. The covariance generated by our theory agrees with the observed structure.

**Corollary 2.3.** *Assume the update is equivariant at all steps. Given Assumption 2.1 and Theorem 2.2, by induction, the distribution is invariant at all steps, $p_t(G\boldsymbol{\theta}_t) = p_t(\boldsymbol{\theta}_t), \forall t$.*

*Remark.* This result does not help in finding the functional form of the distribution, that remains unknown except in very simple cases (see Appendix B for a toy example where the distribution remains Gaussian at all time steps).

### 2.4. Constraints on mean and covariance of gradient

In this section, we demonstrate that the invariance of the loss function, as outlined in Section 2.2, together with the invariance of the probability distribution, discussed in Section 2.3, imposes specific constraints on both the gradient mean and covariance. Furthermore, we note that any transformation $G$ belonging to one of the three groups $\mathbb{G}_o, \mathbb{G}_b, \mathbb{G}_s$ is orthogonal, meaning that all its elements satisfy the property $G^T = G^{-1}$.

**Lemma 2.4.** *Assume both the loss function and the probability distribution are invariant for an orthogonal transformation $G$, namely $\mathcal{L}(G\boldsymbol{\theta}_t) = \mathcal{L}(\boldsymbol{\theta}_t)$ and $p_t(G\boldsymbol{\theta}_t) = p_t(\boldsymbol{\theta}_t)$. Assume that the mean $\boldsymbol{\mu}$ and covariance $\Sigma$ of the gradient exist and are finite. Then, the mean satisfies the eigenvalue equation*

$$\boldsymbol{\mu}_t = G\boldsymbol{\mu}_t \tag{13}$$

*and the covariance matrix is invariant upon the congruent transformation*

$$\Sigma_t = G\Sigma_t G^T \tag{14}$$

*Furthermore, any analytic matrix function $f(\Sigma)$ of the covariance matrix satisfies the same equation*

$$f(\Sigma_t) = Gf(\Sigma_t)G^T \tag{15}$$

The proof is provided in Appendix G. It is important to note that Equations (13) and (14) are linear and homogeneous,

which implies the possibility of infinitely many solutions (see Lemma C.3). In Section 3, we identify a solution space that remains valid for all members of a given symmetry group. Since this space is infinite, both the mean $\boldsymbol{\mu}_t$ and the covariance $\Sigma_t$ may evolve to different values at different time steps, while still satisfying Equations (13) and (14).

Appendix H demonstrates that Equations (14) and (15) hold identically for the Hessian matrix when averaged over the parameter distribution. Additionally, it provides examples of Hessians observed in experiments (Figures 7, 8, 9). Similar results extend to other types of curvature matrices, including the Fisher Information matrix and the Gauss-Newton matrix.

## 3. Results

In this section, we derive exact expressions for the mean and covariance of the gradient in a MLP and compare them with empirical results obtained from experiments on synthetic data. Given the layered architecture of MLPs and the distinction between weights and biases, we introduce a flattened vector of parameters encompassing all layers, by concatenating and vectorizing all tensors in column-major order, given by

$$\boldsymbol{\theta} = \mathrm{Vec}\left(W_1, \mathbf{b}_1, \ldots, W_L, \mathbf{b}_L\right) \tag{16}$$

We visualize the gradient covariance matrix by vectorizing the parameters according to Equation (16). For the Hessian matrix, we present analogous results in Figures 7, 8, and 9 of Appendix H.

Figure 2 presents the gradient covariance matrix for a two-layer MLP with Tanh activations in a tiny model with five neurons per layer. The results are shown for both empirical experiments (panels A and B, see Section 4 for details) and our theoretical predictions (panel C, see Section 3.2 for details). The structure observed in experiments closely aligns with theoretical predictions, both before training (panel A,

$t = 0$) and after 20 steps of gradient descent (panel B, $t = 20$). We highlight that our theoretical framework captures the overall structure of the covariance matrix rather than its exact numerical values. The estimation of specific values is further discussed in Section 3.4.

Figure 2 further reveals that input weights exhibit column-wise correlations, likely reflecting dependencies in their input data, while output weights show row-wise correlations, capturing similarities in their output. Additionally, correlations are observed between the rows of the output weights and the columns of the input weights, potentially indicating input-output dependencies. Figure 3 extends this analysis to a three-layer MLP. The observed gradient covariance structure remains consistent with theoretical predictions. However, the covariance matrix becomes block-diagonal, indicating a significant reduction in inter-layer correlations as the number of layers increases from two to three. As demonstrated in Section 3.2, our theoretical framework predicts that this absence of correlations between layers persists for architectures with more than three layers. We speculate that the invariance with respect to sign changes of both the incoming and outgoing weights of a neuron may lead to cancellations in the covariance of these weights. However, further studies are needed to understand this observation.

Figure 4 presents the gradient covariance matrix for a three-layer MLP with ReLU activations and excluding biases. Once again, our theoretical framework accurately captures the observed structure of the covariance matrix, both before training and after a few optimization steps. Notably, the reduced symmetry of ReLU activations results in a more intricate covariance structure, characterized by significant correlations across layers. As demonstrated in Section 3.3, these inter-layer correlations persist regardless of the network's depth. In the following sections, we provide the theoretical framework used to derive the gradient covariance structures observed in Figures 2, 3, and 4.

### 3.1. Theoretical results

In this section, we present solutions to Equations (13) and (14) that remain valid for all possible transformations $G$ within a given symmetry group. These solutions are derived for each of the three symmetry groups introduced in Section 2.2. Using the vectorization of parameters of Equation (16), the transformation in Equations (6), (7) can be rewritten in block-diagonal form

$$G = \begin{pmatrix} V_0 \otimes V_1 & & & & \\ & V_1 & & & \\ & & \ddots & & \\ & & & V_{L-1} \otimes V_L & \\ & & & & V_L \end{pmatrix} \quad (17)$$

where $\otimes$ denotes Kronecker product. For the mean gradient, we use a notation similar to the parameter vector, Equation (16)

$$\boldsymbol{\mu} = \mathbb{E}_{\boldsymbol{\theta}} \operatorname{Vec}\left(\frac{\partial \mathcal{L}}{\partial W_1}, \frac{\partial \mathcal{L}}{\partial \mathbf{b}_1}, \ldots, \frac{\partial \mathcal{L}}{\partial W_L}, \frac{\partial \mathcal{L}}{\partial \mathbf{b}_L}\right) = \quad (18)$$
$$= \operatorname{Vec}\left(\boldsymbol{\mu}_1, \tilde{\boldsymbol{\mu}}_1, \ldots, \boldsymbol{\mu}_L, \tilde{\boldsymbol{\mu}}_L\right) \quad (19)$$

where we define $\boldsymbol{\mu}_\ell = \mathbb{E}_{\boldsymbol{\theta}} \frac{\partial \mathcal{L}}{\partial \mathbf{w}_\ell}$, $\tilde{\boldsymbol{\mu}}_\ell = \mathbb{E}_{\boldsymbol{\theta}} \frac{\partial \mathcal{L}}{\partial \mathbf{b}_\ell}$, and $\mathbf{w}_\ell = \operatorname{Vec}(W_\ell)$. Similarly, we define $\Sigma_{\ell\ell'} = \mathbb{E}_{\boldsymbol{\theta}} \frac{\partial \mathcal{L}}{\partial \mathbf{w}_\ell} \frac{\partial \mathcal{L}}{\partial \mathbf{w}_{\ell'}}^T$, $\tilde{\Sigma}_{\ell\ell'} = \mathbb{E}_{\boldsymbol{\theta}} \frac{\partial \mathcal{L}}{\partial \mathbf{b}_\ell} \frac{\partial \mathcal{L}}{\partial \mathbf{w}_{\ell'}}^T$ and $\tilde{\tilde{\Sigma}}_{\ell\ell'} = \mathbb{E}_{\boldsymbol{\theta}} \frac{\partial \mathcal{L}}{\partial \mathbf{b}_\ell} \frac{\partial \mathcal{L}}{\partial \mathbf{b}_{\ell'}}^T$. Then, the covariance of the gradient is written in terms of its constituent blocks as

$$\Sigma = \begin{pmatrix} \Sigma_{11} & \tilde{\Sigma}_{11}^T & \ldots & \Sigma_{1L} & \tilde{\Sigma}_{L1}^T \\ \tilde{\Sigma}_{11} & \tilde{\tilde{\Sigma}}_{11} & \ldots & \tilde{\Sigma}_{1L} & \tilde{\tilde{\Sigma}}_{LL} \\ \vdots & \vdots & \ddots & \vdots & \vdots \\ \Sigma_{L1} & \tilde{\Sigma}_{1L}^T & \ldots & \Sigma_{LL} & \tilde{\Sigma}_{LL}^T \\ \tilde{\Sigma}_{L1} & \tilde{\tilde{\Sigma}}_{L1} & \ldots & \tilde{\Sigma}_{LL} & \tilde{\tilde{\Sigma}}_{LL} \end{pmatrix} \quad (20)$$

Then, Equation (13) for the mean gradient becomes

$$\boldsymbol{\mu}_\ell = (V_{\ell-1} \otimes V_\ell) \boldsymbol{\mu}_\ell \quad (21)$$
$$\tilde{\boldsymbol{\mu}}_\ell = V_\ell \tilde{\boldsymbol{\mu}}_\ell \quad (22)$$

for all $\ell = 1, \ldots, L$. Equation (14) for the covariance of the gradient becomes

$$\Sigma_{\ell\ell'} = (V_{\ell-1} \otimes V_\ell) \Sigma_{\ell\ell'} \left(V_{\ell'-1}^T \otimes V_{\ell'}^T\right) \quad (23)$$
$$\tilde{\tilde{\Sigma}}_{\ell\ell'} = V_\ell \tilde{\tilde{\Sigma}}_{\ell\ell'} V_{\ell'}^T \quad (24)$$
$$\tilde{\Sigma}_{\ell\ell'} = V_\ell \tilde{\Sigma}_{\ell\ell'} \left(V_{\ell'-1}^T \otimes V_{\ell'}^T\right) \quad (25)$$

for all $\ell, \ell' = 1, \ldots, L$. In the next sections, we provide solutions to Equations (21), (22), (23), (24), (25).

### 3.2. Odd activations (e.g. linear or Tanh)

Here, we examine both linear and nonlinear odd activations (e.g., Tanh) together, as they yield the same structure for the mean and covariance of the gradient.

**Theorem 3.1.** *Assume $L > 1$. If the loss and distribution of parameters are invariant for either groups $\mathbb{G}_o$, $\mathbb{G}_b$, then the mean gradient is equal to*

$$\boldsymbol{\mu}_\ell = \mathbf{0} \qquad \text{for } \ell = 1, \ldots, L \quad (26)$$
$$\tilde{\boldsymbol{\mu}}_\ell = \mathbf{0} \qquad \text{for } \ell = 1, \ldots, L-1 \quad (27)$$
$$\tilde{\boldsymbol{\mu}}_L = \tilde{\mathbf{z}}_L \quad (28)$$

*where $\tilde{\mathbf{z}}_L$ is a vector of size $d_L$. The covariance of the*

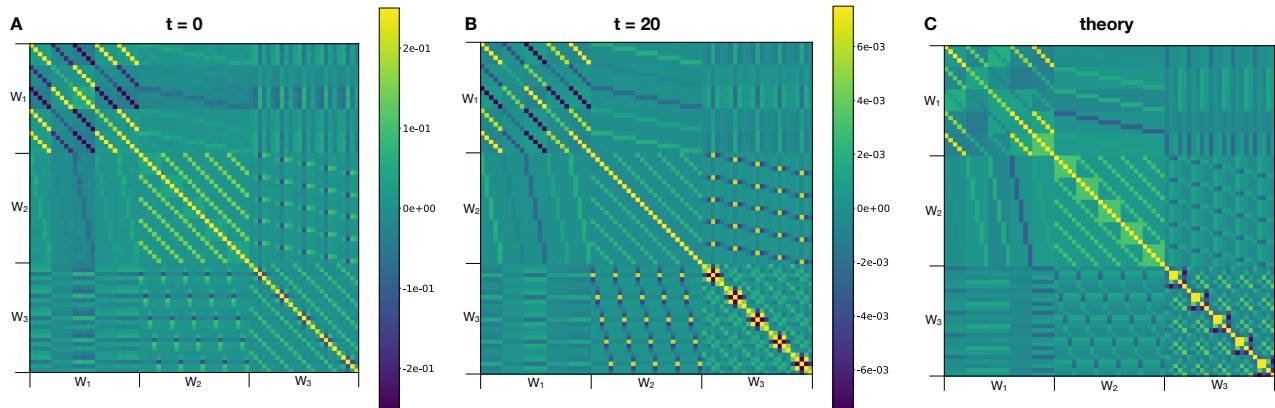

*Figure 4.* **Gradient covariance matrix for three-layer MLP with ReLU activations**. Similar to Figure 3 but without bias and averaging over 100000 models. In contrast to Tanh activations, ReLU activations retain correlations across layers. The covariance shows a remarkably complex structure that is well predicted by theory.

*gradient, given in terms of its constituent blocks, is equal to*

$$\Sigma_{11} = \Phi_1 \otimes \mathrm{I}_{d_1} \tag{29}$$

$$\Sigma_{\ell\ell} = \phi_\ell \left(\mathrm{I}_{d_{\ell-1}} \otimes \mathrm{I}_{d_\ell}\right) \qquad \text{for } \ell = 2, \ldots, L-1 \tag{30}$$

$$\Sigma_{LL} = \mathrm{I}_{d_{L-1}} \otimes \Phi_L \tag{31}$$

$$\Sigma_{12} = \left(\Psi_1 \otimes \mathrm{I}_{d_1}\right) K \qquad \text{if } L = 2 \tag{32}$$

$$\tilde{\tilde{\Sigma}}_{\ell\ell} = \tilde{\tilde{\phi}}_\ell \, \mathrm{I}_{d_\ell} \qquad \text{for } \ell = 1, \ldots, L-1 \tag{33}$$

$$\tilde{\tilde{\Sigma}}_{LL} = \tilde{\tilde{\Phi}}_L \tag{34}$$

$$\tilde{\Sigma}_{11} = \tilde{\phi}_1^T \otimes \mathrm{I}_{d_1} \tag{35}$$

$$\tilde{\Sigma}_{L-1,L} = \mathrm{I}_{d_{L-1}} \otimes \tilde{\phi}_{L-1}^T \tag{36}$$

*where $\phi_\ell$, $\tilde{\tilde{\phi}}_\ell$ are positive scalars, $\tilde{\phi}_1 \in \mathbb{R}^{d_0}$, $\tilde{\phi}_{L-1} \in \mathbb{R}^{d_L}$ are column vectors, $\Phi_1 \in \mathbb{R}^{d_0 \times d_0}$, $\Phi_L \in \mathbb{R}^{d_L \times d_L}$, $\tilde{\tilde{\Phi}}_L \in \mathbb{R}^{d_L \times d_L}$, are positive-definite matrices and $\Psi_1 \in \mathbb{R}^{d_0 \times d_2}$ is a matrix. All other terms are zero. $K$ is the commutation matrix (see Chapter 3.7 of Magnus & Neudecker (2019)).*

Proof is in Appendix I. We refer to the set $(\tilde{\mathbf{z}}_L, \phi_\ell, \tilde{\tilde{\phi}}_\ell, \tilde{\phi}_1, \tilde{\phi}_{L-1}, \Phi_1, \Phi_L, \tilde{\tilde{\Phi}}_L, \Psi_1)$ as the factors, which are undetermined. We provide an algorithm for estimating the factors in Section 3.4 The number of undetermined factors is of order $\mathcal{O}(d^2 + L)$. We note that the values of those factors should ensure that the covariance is positive semi-definite.

### 3.3. Other activations (ReLU)

In the case of non-symmetric activations, we have the following

**Theorem 3.2.** *Assume $L > 1$. If the loss and distribution of parameters are invariant for the group $\mathbb{G}_s$, then the mean gradient is described by Equations (113)-(117) in Appendix J. The covariance of the gradient, given in terms of its*

*constituent blocks, is described by Equations (118)-(142) in Appendix J.*

The proof is also provided in Appendix J. We note that the number of undetermined factors is $\mathcal{O}(d^2 + L^2)$. The number of factors is larger than the case of odd activations, thus revealing a more complex structure. We interpret this result as a consequence of the smaller size of the group $\mathbb{G}_s$ with respect to $\mathbb{G}_o$, $\mathbb{G}_b$ (see Section 2.2).

### 3.4. A practical algorithm for second-order optimization

We observed in Figures 2,3,4, that the covariance is very structured. In contrast to previous approximations (Martens & Grosse, 2015; Eschenhagen et al., 2024), the matrix is not block-diagonal, although it is Kronecker-factorized. This structure may capture the most important variations in the loss landscape, therefore we use it for computing the second-order update in Equation (1). We break down the computation of the second-order update into three steps: 1) Estimate the factors of the covariance; 2) Compute the factors of its inverse square root; 3) Compute the matrix-vector product between the inverse square root of the covariance and the gradient. We demonstrate that all three steps can be performed efficiently, leveraging the convenient properties of the matrix. A detailed description of the complete procedure is provided in Algorithm 1 in the Appendix, using the simple case of a two-layer MLP with Tanh activation and no bias. Similar steps can be applied to other cases.

The gradient covariance, as defined in Equation (3), requires averaging over models initialized with different parameters. However, we observe that the number of factors of the covariance is much smaller than the total number of elements in the covariance matrix. As shown in Figures 2, 3, and 4, many elements of the covariance matrix share identical values. We leverage this observation to estimate

the covariance efficiently, even from the gradient of a *single* model, by averaging over the corresponding pairs of parameters. An example of this estimation process is provided for the case of a two-layer MLP with Tanh activations and no bias. Similar equations can be derived for the case with bias and ReLU activations. We estimate the factors using the following equations:

$$(\Phi_1)_{jl} = \frac{1}{d_1} \sum_{i=1}^{d_1} \left( \frac{\partial \mathcal{L}}{\partial W_1} \right)_{ij} \left( \frac{\partial \mathcal{L}}{\partial W_1} \right)_{il} \tag{37}$$

$$(\Phi_2)_{ik} = \frac{1}{d_1} \sum_{j=1}^{d_1} \left( \frac{\partial \mathcal{L}}{\partial W_2} \right)_{ij} \left( \frac{\partial \mathcal{L}}{\partial W_2} \right)_{kj} \tag{38}$$

$$(\Psi_1)_{jk} = \frac{1}{d_1} \sum_{i=1}^{d_1} \left( \frac{\partial \mathcal{L}}{\partial W_1} \right)_{ij} \left( \frac{\partial \mathcal{L}}{\partial W_2} \right)_{ki} \tag{39}$$

These are matrix-matrix products of size equal to the neural network width, that can be computed efficiently using a GPU.

To quantify the error in estimating the factors of global covariance by a single model, we computed their correlation with a high-precision reference estimate. This reference was obtained by averaging across an ensemble of 10000 models, which serves as our ground truth. Table 1 shows that the correlation between the single-model estimate and the ground truth increases with layer width $d_1$, for all factors $\Phi_1$, $\Psi_2$, $\Phi_2$. To further improve the estimates and reduce error, we apply momentum to Equations (37), (38), and (39) across training iterations. Appendix K provides additional details, along with an empirical analysis of the estimation error in presence of momentum (see Figure 10).

*Table 1.* Correlation between single-model estimates of the factors and ground truth, varying the width of the hidden layer. Single model estimates improve with the layer width.

| Layer widths $(d_0, d_1, d_2)$ | $\Phi_1$ | $\Psi_1$ | $\Phi_2$ |
|---|---|---|---|
| (100, 10, 100) | 0.67±0.05 | 0.38±0.06 | 0.30±0.05 |
| (100, 100, 100) | 0.90±0.02 | 0.61±0.04 | 0.52±0.03 |
| (100, 1000, 100) | 0.96±0.01 | 0.64±0.03 | 0.54±0.03 |
| (100, 10000, 100) | 0.97±0.01 | 0.65±0.03 | 0.56±0.03 |

We denote the flattened vector of all factors by $\phi$. After obtaining estimates of the factors $\phi$ by Equations (37), (38), (39), we could construct the covariance matrix $\Sigma$ using Theorem 3.1, and then compute its inverse square root $\Sigma^{-\frac{1}{2}}$. However, this process becomes computationally expensive for high-dimensional models. To make this step

more feasible, we leverage the fact that any analytic function of the covariance matrix retains the same structure as the covariance itself, as shown by Lemma 2.4 and Equation (15). Therefore, the inverse square root $\Sigma^{-\frac{1}{2}}$ can be fully described by another set of factors, denoted by the vector $\phi^{\text{isr}}$. Instead of computing the full matrix $\Sigma$ and its inverse square root $\Sigma^{-\frac{1}{2}}$, we compute the factors $\phi^{\text{isr}}$ from $\phi$, without ever constructing the large matrices. Appendix L provides a detailed procedure for efficiently computing $\phi^{\text{isr}}$. In short, because the factors do not depend on the layer widths $d_1, \ldots, d_{L-1}$, we compute a *surrogate* covariance $\Lambda$ where the layer widths are set to the smallest values allowed ($d_\ell = 1$ for Tanh activations and $d_\ell = 2$ for ReLU activations). The surrogate matrix $\Lambda$ is significantly smaller than the full matrix $\Sigma$. For an MLP, the surrogate matrix has $\mathcal{O}(d_0^2 + d_L^2 + L^2)$ elements, while the full matrix has $\left[ \sum_\ell d_\ell (d_{\ell-1} + 1) \right]^2$ elements. We then compute the inverse square root $\Lambda^{-\frac{1}{2}}$ of the surrogate and derive the corresponding factors $\phi^{\text{isr}}$. Appendix L provides examples comparing the full and surrogate matrices for small MLPs with Tanh or ReLU activations (Figures 11, 12, 13).

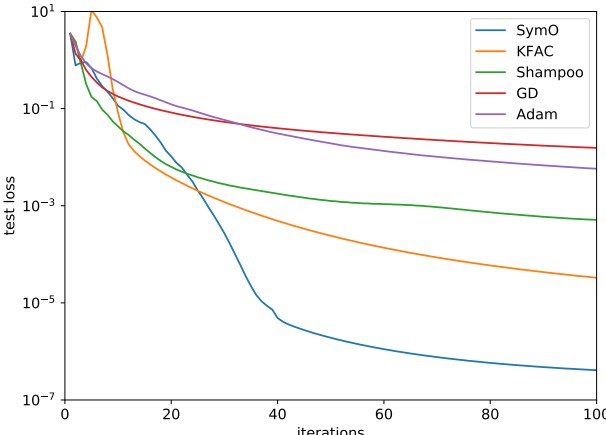

*Figure 5.* **Optimization of a two-layer MLP with linear activations**. Test loss vs training iterations. The Symmetry-based Optimizer (SymO) is compared with other first- and second-order optimizers.

In the final step, we compute the product between the inverse square root of the covariance matrix $\Sigma^{-\frac{1}{2}}$ and the gradient vector, as described in Equation (1). This computation can be carried out efficiently by using the estimated factors $\phi^{\text{isr}}$, without the need to explicitly compute $\Sigma^{-\frac{1}{2}}$. We observe that each block of the covariance matrix (as well as its inverse square root) consists of Kronecker products, which enables efficient computation of matrix-vector products (see e.g. Martens & Grosse (2015)). For the case of a two-layer MLP with Tanh activations and no bias, the update is equal

to (see Equation (206) in Appendix L)

$$(W_1)_{t+1} = (W_1)_t - \alpha \left( \frac{\partial \mathcal{L}}{\partial W_1} \Phi_1^{\text{isr}} + \frac{\partial \mathcal{L}}{\partial W_2^T} \Psi_1^{\text{isr}\,T} \right) \tag{40}$$

$$(W_2)_{t+1} = (W_2)_t - \alpha \left( \Psi_1^{\text{isr}\,T} \frac{\partial \mathcal{L}}{\partial W_1^T} + \Phi_2^{\text{isr}} \frac{\partial \mathcal{L}}{\partial W_2} \right) \tag{41}$$

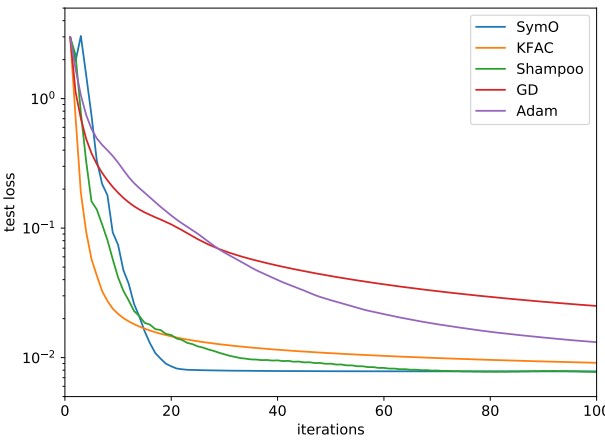

*Figure 6.* **Optimization of a two-layer MLP with Tanh activations**. Test loss vs training iterations. The Symmetry-based Optimizer (SymO) is compared with other first and second order optimizers.

## 4. Experiments

We test second-order optimization on a two-layer MLP without bias and synthetic data, using either linear or Tanh activation. The model has layer width $d_0 = 100$, $d_1 = 70$, $d_2 = 40$, with a total of 9800 parameters. The synthetic dataset consists of 5000 training and 5000 testing data points, where the input is sampled from a Gaussian distribution with zero mean. The covariance matrix of the input is generated using random orthogonal eigenvectors (Mezzadri, 2007), and the eigenvalues are set on a logarithmic grid between $10^{-5}$ and $10^0$. We optimize the square loss, where the target output is provided by a neural network (teacher) with identical architecture of the network to be optimized (student). The weights of both the teacher and student networks are drawn randomly from a Gaussian distribution. We use full-batch optimization, where the gradient is computed over the entire training dataset in each iteration. Although full batch training is uncommon in neural networks, second-order optimizers usually benefit from large batches (Zhang et al., 2019; Anil et al., 2021), therefore we leave the study of mini-batch training to future work.

We compare five optimization algorithms: gradient descent (GD), Adam (Kingma & Ba, 2014), KFAC (Martens &

Grosse, 2015), Shampoo (Gupta et al., 2018) and our optimizer, which we call SymO (Symmetry-based Optimizer). For all optimizers, learning rate is set by a grid search. For second-order optimizers, we additionally set a second hyperparameter by grid search: damping $\lambda$ for KFAC, initialization $\epsilon$ for Shampoo and decay parameter $\beta$ for SymO. See Table 2 for the hyperparameter values used and Appendix M for details. Figures 5, 6 show the optimization trajectories of the five optimizers for, respectively, linear and Tanh activations. The symmetry-based optimizer SymO tends to converge faster than all other optimizers with nearly identical time per iteration.

## 5. Discussion

We introduce a novel theoretical framework for deriving the exact structure of the global curvature of neural networks, and for estimating a second-order optimizer. We provide preliminary evidence suggesting that the structure of the global curvature enhances convergence when applied as a preconditioner. However, a comprehensive analysis of the errors introduced by our approximation will be a subject of future investigation. While we focus on the case of MLP and synthetic data, our framework is general and can be extended to other neural network architectures, including residual networks, convolutional networks, recurrent networks, and transformers. Future work will explore the symmetries and corresponding curvature structures for these architectures, and will evaluate the effectiveness of our approach on real-world data and larger models.

We note that most prior studies on second-order optimization have relied on block-diagonal and Kronecker-factored approximations of the curvature matrix (Martens & Grosse, 2015), which are exact only in models that are either linear in the input (Bernacchia et al., 2018) or in the parameters (Karakida & Osawa, 2020). Our work demonstrates that the global curvature matrix is not inherently block-diagonal, but we do find that matrix blocks exhibit a Kronecker-factored structure. We also note that the block-diagonal terms of the update in Equations (40), (41) are similar to the update of Shampoo (Gupta et al., 2018). Thus, our findings partially justify the approximations used in previous studies and offer a more nuanced understanding of their validity.

Our framework could have significant applications in fields that require second-order estimates for large models. For example, Bayesian deep learning relies on approximating the posterior over parameters by a Gaussian distribution (Blundell et al., 2015; Lin et al., 2019). The covariance of this distribution is usually approximated using a diagonal or block-diagonal structure (Daxberger et al., 2021). Our work offers a method for efficiently computing the full covariance, which may lead to more accurate Bayesian posterior estimates.

## Impact statement

This paper presents work whose goal is to advance the field of Machine Learning. There are many potential societal consequences of our work, none which we feel must be specifically highlighted here.

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

## A. Related work

Numerous efforts have been made to make second-order optimization feasible for deep neural networks. Like our work, prior studies have explored architecture-specific curvature for various models, such as MLPs (Martens & Grosse, 2015), ConvNets (Grosse & Martens, 2016), recurrent networks (Martens et al., 2018), and transformers (Eschenhagen et al., 2024). However, these approaches rely on block-diagonal and kronecker-factored approximations, which lack theoretical guarantees and are exact only in limited scenarios, such as when networks are linear either in their input (Bernacchia et al., 2018) or their parameters (Karakida & Osawa, 2020). Recent research has aimed to compute curvature *exactly*, without relying on block-diagonal approximations, but such methods are restricted to small-scale models (Cai et al., 2019; Arbel et al., 2023; Korbit et al., 2024) or to reversible networks (Buffelli et al., 2024). Crucially, none of these studies leverage the inherent symmetries present in neural network architectures. In contrast, our work introduces a novel framework that derives the *exact* structure of curvature matrices by utilizing model symmetries for the first time. This approach is not limited by model size and can be scaled to accommodate any large neural network with symmetries, providing a robust and widely applicable solution to second-order optimization.

Neural networks exhibit a range of symmetries that can be broadly categorized into discrete (e.g., permutations and sign flips, Chen et al. (1993)) and continuous (e.g., scaling, translations, and rotations, Kunin et al. (2020); Ziyin (2024)). While continuous symmetries have been extensively studied for their potential to improve optimization (Neyshabur et al., 2015; Meng et al., 2018; Zhao et al., 2022; 2023b), discrete symmetries have received comparatively less attention. Our work is the first to leverage discrete symmetries specifically to enhance optimization. Prior research has employed discrete symmetries in other contexts, such as model merging (Tatro et al., 2020; Entezari et al., 2021; Ainsworth et al., 2022; Jordan et al., 2023; Navon et al., 2024) and analyzing the structure and stability of fixed points in the loss landscape (Fukumizu & Amari, 2000; Simsek et al., 2021; Zhang et al., 2021). Despite these applications, their direct use in optimization strategies remains unexplored, and our work fills this critical gap. Additionally, the study of optimization dynamics has incorporated both discrete (Chen et al., 2024; Ziyin, 2024) and continuous symmetries (Du et al., 2018; Kunin et al., 2020; Zhao et al., 2023a; Ziyin et al., 2024), highlighting the broader interest in understanding how symmetries influence training. By focusing on how discrete symmetries may improve optimization, we introduce a novel perspective that complements existing approaches.

A key distinction of our approach compared to prior work on second-order optimization and symmetries is our focus on global rather than local curvature, also termed position-independent versus position-dependent curvature (Titsias, 2024). Existing studies predominantly estimate local curvature, which depends on a specific set of model parameters and captures the loss landscape's behavior around a single point. Only two previous works, Yang et al. (2021) and Yao et al. (2021), have explored global curvature, though with different interpretations. Yang et al. (2021) characterize global structure through mode connectivity (Garipov et al., 2018), while Yao et al. (2021) define it via an exponential moving average of curvature. Instead, we define global curvature by an average of the curvature matrix over an ensemble of model parameters, and we show that it can be estimated efficiently.

Similar to our study, Kunin et al. (2020) explores symmetry-induced constraints on curvature. However, their work is restricted to analyzing local curvature, which limits them to continuous symmetries and yields fewer constraints compared to the stronger and more numerous constraints that emerge when analyzing the effect of discrete symmetries on global curvature. Furthermore, global curvature estimation can offer more robust insights for optimization, as it is not tied to a specific parameter configuration. The only previous study to incorporate *global* curvature is Titsias (2024), which investigates position-independent preconditioning. However, their approach does not integrate symmetries into the computation, whereas our approach uniquely combines global curvature estimation with symmetry constraints.

## B. Quadratic loss

In this section, we study a toy example to build some intuition into the dynamics of parameters when optimizing a loss function using the update introduced in Section 2.1, Equation (1). Given a parameter vector $\boldsymbol{\theta} \in \mathbb{R}^p$, a constant and positive definite Hessian matrix $H \in \mathbb{R}^{p \times p}$, we consider the quadratic loss function

$$\mathcal{L}(\boldsymbol{\theta}) = \frac{1}{2}(\boldsymbol{\theta} - \boldsymbol{\theta}^*)^T H (\boldsymbol{\theta} - \boldsymbol{\theta}^*) \tag{42}$$

The global minimum of this loss is $\boldsymbol{\theta}^*$. When optimizing this loss by gradient descent, the path taken in the space of parameters depends strongly on the Hessian matrix $H$. If $H$ is a scalar matrix, $H \propto \mathrm{I}$, then the loss is isotropic and the parameters follow a straight line to the minimum. Instead, if $H$ is badly conditioned, then some directions in parameter

space are much more curved than others, and gradient descent takes a convoluted path, which may significantly slow down convergence. Second-order optimization straightens the path and accelerates convergence.

The gradient of the quadratic loss is equal to $\nabla \mathcal{L} = H(\boldsymbol{\theta} - \boldsymbol{\theta}^*)$. We are given an ensemble of models following a normal distribution $p(\boldsymbol{\theta}) = \mathcal{N}(\boldsymbol{\theta}|\boldsymbol{\mu_\theta}, \Sigma_{\boldsymbol{\theta}})$, where $\boldsymbol{\mu_\theta}$ and $\Sigma_{\boldsymbol{\theta}}$ are, respectively, the mean and covariance of the parameters. Since the gradients are a linear function of the parameters, gradients also follow a normal distribution, $\nabla \mathcal{L} \sim \mathcal{N}(\boldsymbol{\mu_g}, \Sigma_{\mathbf{g}})$, where the mean and covariance are given by

$$\boldsymbol{\mu_g} = H(\boldsymbol{\mu_\theta} - \boldsymbol{\theta}^*) \quad \text{and} \quad \Sigma_{\mathbf{g}} = H\Sigma_{\boldsymbol{\theta}}H \tag{43}$$

We define the optimization dynamics by the continuous flow

$$\frac{d\boldsymbol{\theta}}{dt} = -\alpha\Sigma_{\mathbf{g}}^\gamma\nabla\mathcal{L} = -\alpha\Sigma_{\mathbf{g}}^\gamma H(\boldsymbol{\theta} - \boldsymbol{\theta}^*) \tag{44}$$

This is the continuous-time version of the second-order optimization update considered in Section 2.1, Equation (1), with $\gamma = -\frac{1}{2}$. In our derivation, we keep a generic value of $\gamma$ and we later argue in favor of the choice $\gamma = -\frac{1}{2}$.

Since the update of Equation (44) is linear in the parameters, the distribution of parameters remains Gaussian at all time steps, and so does the distribution of gradients. However, the mean and covariance of both distributions change in time. The dynamics of the mean is obtained by averaging Equation (44), while the dynamics of the covariance is obtained by averaging the same equation multiplied by the parameter vector. We find

$$\frac{d\boldsymbol{\mu_\theta}}{dt} = -\alpha\Sigma_{\mathbf{g}}^\gamma H(\boldsymbol{\mu_\theta} - \boldsymbol{\theta}^*) \tag{45}$$

$$\frac{d\Sigma_{\boldsymbol{\theta}}}{dt} = -\alpha\left(\Sigma_{\mathbf{g}}^\gamma H\Sigma_{\boldsymbol{\theta}} + \Sigma_{\boldsymbol{\theta}} H\Sigma_{\mathbf{g}}^\gamma\right) \tag{46}$$

Using the identity $\Sigma_{\mathbf{g}} = H\Sigma_{\boldsymbol{\theta}}H$, and assuming that $H$ and $\Sigma_{\boldsymbol{\theta}}$ commute, we obtain

$$\frac{d\boldsymbol{\mu_\theta}}{dt} = -\alpha\, H^{1+2\gamma}\, \Sigma_{\boldsymbol{\theta}}^\gamma(\boldsymbol{\mu_\theta} - \boldsymbol{\theta}^*) \tag{47}$$

$$\frac{d\Sigma_{\boldsymbol{\theta}}}{dt} = -2\alpha\, H^{1+2\gamma}\, \Sigma_{\boldsymbol{\theta}}^{1+\gamma} \tag{48}$$

We note that these equations simplify by setting $\gamma = -\frac{1}{2}$, because the dependence on $H$ disappears. In that case, the equation for the covariance becomes $\frac{d\Sigma_{\boldsymbol{\theta}}}{dt} = -2\alpha\,\Sigma_{\boldsymbol{\theta}}^{1/2}$. We further assume an isotropic initialization of parameters, $\Sigma_{\boldsymbol{\theta}} \propto \mathrm{I}$ at $t = 0$. The isotropic initialization is a natural choice in absence of any structured prior. In that case, the distribution remains isotropic at all times, $\Sigma_{\boldsymbol{\theta}} = s\mathrm{I}$, where the scalar $s$ is time-dependent. This also ensures commutation of $\Sigma_{\boldsymbol{\theta}}$ and $H$. Then, the optimization dynamics is described by

$$\frac{d\boldsymbol{\mu_\theta}}{dt} = -\frac{\alpha}{\sqrt{s}}(\boldsymbol{\mu_\theta} - \boldsymbol{\theta}^*) \tag{49}$$

$$\frac{ds}{dt} = -2\alpha\sqrt{s} \tag{50}$$

In this case, the mean of the distribution takes a straight path to the minimum $\boldsymbol{\theta}^*$. When considering optimization in discrete time, this implies that a large learning rate can be used and convergence is faster.

For a constant learning rate $\alpha$, we show that the continuous dynamics converge in finite time. The solutions of Equations (49),(50) are

$$\boldsymbol{\mu_\theta} = \left(1 - \frac{\alpha t}{\sqrt{s_0}}\right)(\boldsymbol{\mu_0} - \boldsymbol{\theta}^*) + \boldsymbol{\theta}^* \tag{51}$$

$$s = (\sqrt{s_0} - \alpha t)^2 \tag{52}$$

where $\boldsymbol{\mu}_0$ and $s_0$ are the initial values at $t = 0$, and the dynamics stops at $t = \frac{\sqrt{s_0}}{\alpha}$, when all models have converged to the minimum $\boldsymbol{\theta}^*$. We note that discretization of the dynamics in Equations (49),(50) has a large error near convergence, even if the learning rate $\alpha$ is small. Therefore, we expect that discrete parameter updates would differ significantly from the predicted solution in Equations (51),(52). To avoid this problem, we consider an exponentially decaying learning rate

$$\alpha = \alpha_0 \beta^t \tag{53}$$

with $0 < \beta < 1$. With this choice of the learning rate, and assuming that $\alpha_0$ and $\beta$ are chosen to satisfy $\sqrt{s_0} = \frac{\alpha_0}{\log(\frac{1}{\beta})}$, the solutions of Equations (49),(50) are equal to

$$\boldsymbol{\mu_\theta} = \beta^t (\boldsymbol{\mu}_0 - \boldsymbol{\theta}^*) + \boldsymbol{\theta}^* \tag{54}$$

$$s = \left(\frac{\alpha_0 \beta^t}{\log(\beta)}\right)^2 \tag{55}$$

In this case as well, all models converge quickly (exponentially fast) to the minimum of the loss. With the choice of an exponentially decaying learning rate, the discretization error of the dynamics in Equations (49),(50) remains small, therefore we expect the discrete updates to follow closely the solution in Equations (54),(55). The assumed equality $\sqrt{s_0} = \frac{\alpha_0}{\log(\frac{1}{\beta})}$ is satisfied, for example, when $s_0 \sim 1$, $\alpha_0$ is small and $\beta \sim 1 - \alpha_0$. We find the relationship $\beta \sim 1 - \alpha_0$ to hold in our experiments, see Table 2.

## C. Existence of invariant measures

In this section, we discuss the conditions for a transformation $G$ to accept an invariant probability density and, given a suitable $G$, what can we say about the mean and covariance of the invariant probability. Those conditions are identical to those of the mean and covariance of the loss gradient, discussed in the main text, therefore the results of this section can be used to analyze the properties of the gradient statistics as well. While in the main text we assume that $G$ is orthogonal, the results of this section do not use that assumption and are valid also for non-orthogonal $G$.

A necessary condition for the existence of an invariant probability density is that the transformation is volume-preserving. This can be shown by computing the normalization condition. Denoting the probability density $p(\boldsymbol{\theta})$ and the linear transformation $G$, invariance is given by $p(G\boldsymbol{\theta}) = p(\boldsymbol{\theta})$. The domain of $p$ is also invariant. Using the change of variable $\boldsymbol{\theta} = G\boldsymbol{\theta}'$, $d\boldsymbol{\theta} = |\det(G)|d\boldsymbol{\theta}'$, the normalization condition can be written as

$$1 = \int d\boldsymbol{\theta}\, p(\boldsymbol{\theta}) = |\det(G)| \int d\boldsymbol{\theta}'\, p(G\boldsymbol{\theta}') = |\det(G)| \int d\boldsymbol{\theta}'\, p(\boldsymbol{\theta}') = |\det(G)| \tag{56}$$

Therefore, $|\det(G)|$ must be equal to one, which means that the transformation must be volume-preserving. For example, the distribution $\delta(\boldsymbol{\theta})$, the Dirac delta function, is invariant for any volume-preserving transformation, since $\delta(G\boldsymbol{\theta}) = |\det(G)|^{-1}\delta(\boldsymbol{\theta})$. However, besides volume preservation, the transformation $G$ must satisfy stronger requirements to accept non-trivial invariant distributions.

A useful example is provided by rescaling transformations, in which a parameter $w_1$ is multiplied by a constant $\lambda$ and another parameter $w_2$ is multiplied by its inverse $1/\lambda$. Rescaling is volume-preserving and keeps ReLU neural networks invariant (Neyshabur et al., 2015). However, besides the delta function, it is not possible for any other probability distribution of $w_1$ and $w_2$ to be invariant for rescaling, because it would have to be constant along the one-dimensional curves with constant $w_2 w_1$, which are unbounded.

It was shown in Flytzanis (1977) that, to accept an invariant distribution whose domain spans the entire space, the linear transformation must be similar to an orthogonal transformation. Technically speaking, that is equivalent to the requirement that all eigenvalues of $G$ lie on the unit circle of the complex plane. That condition implies volume-preservation, but is much stronger. However, transformations that are similar to orthogonal transformations are not necessarily orthogonal (see Lemma C.2).

We provide a proof that is valid for the case of distributions with finite and positive definite covariance. The proof includes a procedure for constructing an invariant covariance $\Sigma$, given the transformation $G$. First, Lemma C.1 shows that the covariance of an invariant distribution must be invariant upon a congruent transformation, akin to a Lyapunov equation (Simoncini, 2016). Lemma C.2 provides conditions on the transformation $G$ to accept an invariant distribution, and lemma C.3 shows how to construct the covariance of an invariant distribution.

**Lemma C.1.** *Assume that a probability density is invariant for a linear volume-preserving transformation, $p(G\boldsymbol{\theta}) = p(\boldsymbol{\theta})$, and it has a finite mean and covariance. The mean $\boldsymbol{\mu}$ and covariance $\Sigma$ of the distribution must satisfy the following equations*

$$\boldsymbol{\mu} = G\boldsymbol{\mu} \tag{57}$$

$$\Sigma = G\Sigma G^T \tag{58}$$

We note that the equation for $\boldsymbol{\mu}$ is an eigenvalue equation, while the equation for $\Sigma$ is a congruent transformation, also known as homogeneous discrete-time Lyapunov equation (Simoncini, 2016).

*Proof.* The result can be obtained by a change of variable $\boldsymbol{\theta} = G\boldsymbol{\theta}'$, $d\boldsymbol{\theta} = |\det(G)|d\boldsymbol{\theta}'$, using volume preservation, $|\det(G)| = 1$ and the invariance of the probability density:

$$\boldsymbol{\mu} = \int d\boldsymbol{\theta}\, \boldsymbol{\theta}\, p(\boldsymbol{\theta}) = |\det(G)| \int d\boldsymbol{\theta}'\, G\boldsymbol{\theta}'\, p(G\boldsymbol{\theta}') = G \int d\boldsymbol{\theta}'\, \boldsymbol{\theta}'\, p(\boldsymbol{\theta}') = G\boldsymbol{\mu} \tag{59}$$

$$\Sigma = \int d\boldsymbol{\theta}\, \boldsymbol{\theta}\boldsymbol{\theta}^T p(\boldsymbol{\theta}) - \boldsymbol{\mu}\boldsymbol{\mu}^T = |\det(G)| \int d\boldsymbol{\theta}'\, G\boldsymbol{\theta}'\boldsymbol{\theta}'^T G^T p(G\boldsymbol{\theta}') - G\boldsymbol{\mu}\boldsymbol{\mu}^T G^T = \tag{60}$$

$$= G \left[ \int d\boldsymbol{\theta}'\, \boldsymbol{\theta}'\boldsymbol{\theta}'^T p(\boldsymbol{\theta}') - \boldsymbol{\mu}\boldsymbol{\mu}^T \right] G^T = G\Sigma G^T \tag{61}$$

$\square$

**Lemma C.2.** *If the homogeneous discrete-time Lyapunov equation*

$$\Sigma = G\Sigma G^T \tag{62}$$

*has a positive definite solution $\Sigma$, then $G$ must be similar to an orthogonal matrix, thus it can be diagonalized and its eigenvalues must lie on the unit circle of the complex plane.*

*Proof.* Since $\Sigma$ is positive definite, its square root and inverse exist and are unique. We define the matrix $U$ as

$$U = \Sigma^{-\frac{1}{2}} G \Sigma^{\frac{1}{2}} \quad \Longleftrightarrow \quad G = \Sigma^{\frac{1}{2}} U \Sigma^{-\frac{1}{2}} \tag{63}$$

Substituting this expression into the Lyapunov equation (62), we obtain

$$\Sigma = \Sigma^{\frac{1}{2}} U \Sigma^{-\frac{1}{2}} \Sigma \Sigma^{-\frac{1}{2}} U^T \Sigma^{\frac{1}{2}} = \Sigma^{\frac{1}{2}} U U^T \Sigma^{\frac{1}{2}} \quad \Longleftrightarrow \quad U U^T = \mathrm{I} \tag{64}$$

Therefore, $U$ must be an orthogonal matrix, and $G$ is similar to $U$, therefore it must be diagonalizable and have all eigenvalues in the unit circle of the complex plane.

$\square$

For example, a Gaussian distribution with zero mean and covariance equal to $\Sigma$ is invariant for $G = \Sigma^{\frac{1}{2}} U \Sigma^{-\frac{1}{2}}$, for a fixed $\Sigma$ and any orthogonal $U$. However, other distributions, not necessarily Gaussian, may be invariant for the same group of transformations, for example any distribution with the same level sets. Many other distributions may be invariant only for a small subset of $U$ values, for example permutations, or reflections / rotations with respect to a specific axis.

**Lemma C.3.** *Given a transformation $G$ that is similar to an orthogonal transformation, with real Jordan form given by*

$$G = VBV^{-1} \tag{65}$$

*where $B$ is orthogonal and block-diagonal, with a $+1$ or $-1$ diagonal entry for each real eigenvalue of $G$ and a $2 \times 2$ rotation block for each pair of complex conjugate eigenvalues of $G$. The columns of the real matrix $V$ are the real and imaginary parts of all the right eigenvectors of $G$.*

*Then, a solution $\Sigma$ of the Lyapunov equation (62) is equal to*

$$\Sigma = VDV^T \tag{66}$$

*If $G$ has distinct eigenvalues, the matrix $D$ is diagonal with a $1 \times 1$ block, with an arbitrary positive value, for each real eigenvalue of $G$ (either $+1$ or $-1$), and a $2 \times 2$ diagonal scalar matrix block, with an arbitrary positive value, for each complex conjugate pair of eigenvalues of $G$.*

*If $G$ has degenerate eigenvalues, $D$ is block-diagonal with an arbitrary $\mu \times \mu$ positive definite matrix block for each real eigenvalue of $G$, where $\mu$ is its multiplicity, and a $2\mu \times 2\mu$ block of the form $D_\mu \otimes I_2$ where $D_\mu$ is an arbitrary positive definite matrix of size $\mu \times \mu$, for each complex conjugate pair of eigenvalues of $G$, where $\mu$ is their multiplicity.*

*Proof.* Using the Jordan form (65), we rewrite the Lyapunov equation (62) as

$$\Sigma = VBV^{-1}\Sigma V^{-1^T}B^TV^T \quad \Longleftrightarrow \quad V^{-1}\Sigma V^{-1^T} = BV^{-1}\Sigma V^{-1^T}B^T \tag{67}$$

We define the positive definite matrix

$$D = V^{-1}\Sigma V^{-1^T} \quad \Longleftrightarrow \quad \Sigma = VDV^T \tag{68}$$

then the above equation is rewritten as

$$D = BDB^T \tag{69}$$

Since $B$ is orthogonal, this equation implies that $D$ and $B$ commute. Therefore, for either $+1$ or $-1$ diagonal entries in $B$, the matrix $D$ has a single scalar positive value on the diagonal. For each complex conjugate pair of eigenvalues $e^{\pm i\phi}$, the matrix $B$ has a $2 \times 2$ block

$$\begin{pmatrix} \cos(\phi) & \sin(\phi) \\ -\sin(\phi) & \cos(\phi) \end{pmatrix} \tag{70}$$

The only symmetric matrix that commutes with a 2-dimensional rotation is the identity matrix, therefore $D$ must have a $2 \times 2$ positive scalar matrix block.

The case in which $G$ has degenerate eigenvalues is a straightforward generalization of the case with distinct eigenvalues. In that case: when $B$ has an identity block $I_{\mu_0}$ where $\mu_0$ is the multiplicity of eigenvalue 1, then $D$ has a corresponding block with an arbitrary positive definite matrix $D_{\mu_0}$ of size $\mu_0 \times \mu_0$; when $B$ has a minus identity block $-I_{\mu_\pi}$ where $\mu_\pi$ is the multiplicity of eigenvalue $-1$, then $D$ has a corresponding arbitrary positive definite matrix $D_{\mu_\pi}$ of size $\mu_\pi \times \mu_\pi$; When $B$ has blocks of size $2\mu_\phi \times 2\mu_\phi$

$$I_{\mu_\phi} \otimes \begin{pmatrix} \cos(\phi) & \sin(\phi) \\ -\sin(\phi) & \cos(\phi) \end{pmatrix} \tag{71}$$

where $\mu_\phi$ is the multiplicity of conjugate eigenvalue pairs of angle $\phi$, then $D$ has a corresponding block of size $2\mu_\phi \times 2\mu_\phi$ of the form $D_{\mu_\phi} \otimes I_2$ where $D_{\mu_\phi}$ is an arbitrary positive definite matrix of size $\mu_\phi \times \mu_\phi$,

$\square$

# D. Invariant initialization of parameters

In this section, we discuss which of the most common parameter inizialization routines used in deep learning are consistent with Assumption 2.1. In other words, we ask whether the probability distribution used to initialize parameters is invariant for the given symmetry group. As described in Section 3, the transformation in Equations (6), (7) can be rewritten in terms of a

single vector of parameters encompassing all layers, by concatenating and vectorizing all tensors. Then, the transformation can be written as $\boldsymbol{\theta} \to G\boldsymbol{\theta}$, where the vector of all parameters is equal to

$$\boldsymbol{\theta} = \text{Vec}\left(W_1, \mathbf{b}_1, \ldots, W_L, \mathbf{b}_L\right) \tag{72}$$

and the transformation has a block-diagonal structure

$$G = \begin{pmatrix} V_0 \otimes V_1 & & & & \\ & V_1 & & & \\ & & \ddots & & \\ & & & V_{L-1} \otimes V_L & \\ & & & & V_L \end{pmatrix} \tag{73}$$

In the next sections, we discuss whether different initialization routines are invariant for the transformation $G$, when $G$ belongs to the three different groups considered in this study, $\mathbb{G}_\text{o}, \mathbb{G}_\text{b}, \mathbb{G}_\text{s}$.

### D.1. Gaussian (normal) initialization

In this case, the probability distribution used to initialize parameters is given by the standard multivariate Gaussian

$$p_0(\boldsymbol{\theta}) = (2\pi\sigma^2)^{-\frac{p}{2}} \exp\left(-\frac{|\boldsymbol{\theta}|^2}{2\sigma^2}\right) \tag{74}$$

where $\sigma^2$ is the variance of the distribution. This distribution is invariant for any orthogonal transformation $\boldsymbol{\theta} \to U\boldsymbol{\theta}$, where $U$ is any orthogonal matrix. It is straightforward to verify that the transformation given by Equation (73) is orthogonal, when all $V_\ell$ for $\ell = 1, \ldots, L$ are all orthogonal. Therefore, Gaussian initialization is invariant for all $\mathbb{G}_\text{o}, \mathbb{G}_\text{b}, \mathbb{G}_\text{s}$.

### D.2. Orthogonal initialization

In this case, each tensor in $\{W_1, \mathbf{b}_1, \ldots, W_L, \mathbf{b}_L\}$ is generated according to a uniform distribution in the compact set of orthogonal matrices (Mezzadri, 2007), by a QR decomposition of a random Gaussian matrix (e.g. generated as in Section D.1), and then dropping the excess rows or columns. An additional step of flipping the sign of some rows and columns is necessary for technical reasons, to make the distribution uniform (see Mezzadri (2007)).

We recall that $V_\ell$ are orthogonal for all $\ell = 1, \ldots, L$, thus $V_\ell^T V_\ell = V_\ell V_\ell^T = \text{I}_{d_\ell}$. The transformation $G$ in Equation (73) corresponds to transforming each weight matrix as $W'_\ell = V_\ell W_\ell V_{\ell-1}^T$. When $d_{\ell-1} \leq d_\ell$, then $W_\ell^T W_\ell = \text{I}_{d_{\ell-1}}$ implies

$$W'^T_\ell W'_\ell = V_{\ell-1} W_\ell^T V_\ell^T V_\ell W_\ell V_{\ell-1}^T = V_{\ell-1} W_\ell^T W_\ell V_{\ell-1}^T = V_{\ell-1} V_{\ell-1}^T = \text{I}_{d_{\ell-1}} \tag{75}$$

Similarly, when $d_{\ell-1} \geq d_\ell$, then $W_\ell W_\ell^T = \text{I}_{d_\ell}$ implies

$$W'_\ell W'^T_\ell = V_\ell W_\ell V_{\ell-1}^T V_{\ell-1} W_\ell^T V_\ell^T = V_\ell W_\ell W_\ell^T V_\ell^T = V_\ell V_\ell^T = \text{I}_{d_\ell} \tag{76}$$

Furthermore, the transformation keeps the uniformity of the group (Mezzadri, 2007), therefore the orthogonal initialization is also invariant for all $\mathbb{G}_\text{o}, \mathbb{G}_\text{b}, \mathbb{G}_\text{s}$.

### D.3. Uniform initialization

In this case, each parameter is initialized by drawing a number from a uniform distribution $\mathcal{U}(a, b)$, where $a$ and $b$ are, respectively, the lower and upper bound of the distribution.

It is straightforward to verify that this distribution is not invariant for orthogonal transformations, therefore ruling out the group $\mathbb{G}_\text{o}$. In fact, the domain of the distribution is a hypercube, which is not invariant for rotations. However, any permutation of the parameters leaves the distribution invariant, therefore the distribution is invariant for the group $\mathbb{G}_\text{s}$. If the lower and upper bound are opposite, $a = -b$, then the distribution is also invariant for sign flips and therefore it is invariant for the group $\mathbb{G}_\text{b}$.

## D.4. Sparse initialization

In this case, a random fraction of parameters is set to zero, and the others are initialized following a Gaussian distribution. This distribution is not invariant for orthogonal transformation, because rotations do not preserve sparsity in general. Therefore the group $\mathbb{G}_o$ is again ruled out. However, the distribution is invariant for both permutations and sign flips, and thus for the groups $\mathbb{G}_b$, $\mathbb{G}_s$.

## D.5. Layer-dependent initialization

In some circumstances, different distributions are used to initialize distributions in different layers. For example, layer-dependent variances when using the normal distribution, or layer-dependent lower and upper bounds when using the uniform distribution. All results of previous sections apply to this case as well, because the transformation of Equation (73) is block-diagonal and therefore applies separately to different layers. In other words, parameters of different layers are not mixed in Equation (73).

## E. Proof of Theorem 2.2

By assumption, the Jacobian matrix of $\mathbf{u}$ is nonsingular except on a set of measure zero. Then, we can use the Coarea formula for computing the probability density after the mapping (Krantz & Parks, 2008). Since the dimension of the image and preimage are equal, then the set of preimages is discrete. Therefore, the probability of $\boldsymbol{\theta}_{t+1}$ at step $t+1$ is given by the change of variable formula

$$p_{t+1}(\boldsymbol{\theta}_{t+1}) = \sum_{\boldsymbol{\theta}_t \in \mathcal{U}_t(\boldsymbol{\theta}_{t+1})} p_t(\boldsymbol{\theta}_t) \, | \det\left(J_{\mathbf{u}_t}(\boldsymbol{\theta}_t)\right)|^{-1} \tag{77}$$

where $J_{\mathbf{u}}$ is the Jacobian matrix of $\mathbf{u}$, and $\mathcal{U}_t(\boldsymbol{\theta}_{t+1})$ is the set of pre-images of $\boldsymbol{\theta}_{t+1}$, namely all values $\boldsymbol{\theta}_t$ such that $\mathbf{u}_t(\boldsymbol{\theta}_t) = \boldsymbol{\theta}_{t+1}$. Given the assumption of equivariance, we have that $\mathbf{u}_t(G\boldsymbol{\theta}_t) = G\boldsymbol{\theta}_{t+1}$ and therefore $G\boldsymbol{\theta}_t \in \mathcal{U}_t(G\boldsymbol{\theta}_{t+1})$ for all $\boldsymbol{\theta}_t \in \mathcal{U}_t(\boldsymbol{\theta}_{t+1})$. On the other hand, for all $\tilde{\boldsymbol{\theta}} \in \mathcal{U}_t(G\boldsymbol{\theta}_{t+1})$, since $G$ is invertible and using again equivariance, it must be $\mathbf{u}_t(G^{-1}\tilde{\boldsymbol{\theta}}) = \boldsymbol{\theta}_{t+1}$, and therefore $G^{-1}\tilde{\boldsymbol{\theta}} \in \mathcal{U}_t(\boldsymbol{\theta}_{t+1})$. Those two observations together imply that the elements of $\mathcal{U}_t(G\boldsymbol{\theta}_{t+1})$ are equal to the elements of $\mathcal{U}_t(\boldsymbol{\theta}_{t+1})$ multiplied by $G$. Therefore, we can compute the transformed probability density as

$$p_{t+1}(G\boldsymbol{\theta}_{t+1}) = \sum_{\boldsymbol{\theta}_t \in \mathcal{U}_t(G\boldsymbol{\theta}_{t+1})} p_t(\boldsymbol{\theta}_t) \, | \det\left(J_{\mathbf{u}_t}(\boldsymbol{\theta}_t)\right)|^{-1} = \sum_{\boldsymbol{\theta}_t \in \mathcal{U}_t(\boldsymbol{\theta}_{t+1})} p_t(G\boldsymbol{\theta}_t) \, | \det\left(J_{\mathbf{u}_t}(G\boldsymbol{\theta}_t)\right)|^{-1} \tag{78}$$

We use invariance of the probability at step $t$, namely $p_t(G\boldsymbol{\theta}_t) = p_t(\boldsymbol{\theta}_t)$, and we note that equivariance and chain rule imply $J_{\mathbf{u}_t}(G\boldsymbol{\theta}_t) = J_{\mathbf{u}_t}(\boldsymbol{\theta}_t)G^{-1}$ and therefore $\det(J_{\mathbf{u}_t}(G\boldsymbol{\theta}_t)) = \det(J_{\mathbf{u}_t}(\boldsymbol{\theta}_t))\det(G)^{-1}$. Finally, since $G$ is volume-preserving, $|\det(G)| = 1$, then

$$p_{t+1}(G\boldsymbol{\theta}_{t+1}) = \sum_{\boldsymbol{\theta}_t \in \mathcal{U}_t(\boldsymbol{\theta}_{t+1})} p_t(\boldsymbol{\theta}_t) \, | \det\left(J_{\mathbf{u}_t}(\boldsymbol{\theta}_t)\right)|^{-1} = p_{t+1}(\boldsymbol{\theta}_{t+1}) \tag{79}$$

## F. Equivariance of optimizers

In this section, we ask whether common optimizers used in deep learning are equivariant for the three groups considered in the main text. All results are derived under the assumption that the loss is invariant. We show that both gradient descent (with or without momentum) and the second-order update of Equation (1) are equivariant for all groups $\mathbb{G}_o$, $\mathbb{G}_b$, $\mathbb{G}_s$, while Adam optimizer is equivariant for $\mathbb{G}_b$, $\mathbb{G}_s$, but not $\mathbb{G}_o$.

The gradient descent update is

$$\boldsymbol{\theta}_t = \mathbf{u}(\boldsymbol{\theta}_{t-1}) = \boldsymbol{\theta}_{t-1} - \alpha \nabla \mathcal{L}(\boldsymbol{\theta}_{t-1}) \tag{80}$$

First, we note that the gradient is equivariant when the linear transformation $G$ is orthogonal. Given the loss $\mathcal{L}(\boldsymbol{\theta})$ and the change of variable $\boldsymbol{\theta}' = G\boldsymbol{\theta}$, we define the transformed loss $\tilde{\mathcal{L}}(\boldsymbol{\theta}) = \mathcal{L}(\boldsymbol{\theta}')$. Using the chain rule, we have

$$\nabla \tilde{\mathcal{L}}(\boldsymbol{\theta}) = G^T \nabla \mathcal{L}(\boldsymbol{\theta}') = G^T \nabla \mathcal{L}(G\boldsymbol{\theta}) \quad \Longrightarrow \quad \nabla \mathcal{L}(G\boldsymbol{\theta}) = G^{T^{-1}} \nabla \tilde{\mathcal{L}}(\boldsymbol{\theta}) \tag{81}$$

by assumption, the loss is invariant, $\mathcal{L} = \tilde{\mathcal{L}}$, and $G$ is orthogonal, $G^{T^{-1}} = G$, therefore

$$\nabla \mathcal{L}(G\boldsymbol{\theta}) = G^{T^{-1}} \nabla \mathcal{L}(\boldsymbol{\theta}) = G \, \nabla \mathcal{L}(\boldsymbol{\theta}) \quad \implies \quad \mathbf{u}(G\boldsymbol{\theta}) = G\mathbf{u}(\boldsymbol{\theta}) \tag{82}$$

We note that the gradient is not equivariant in general, however it is equivariant when the transformation $G$ is orthogonal. The extension to momentun is straightforward, although the update depends on all previous values of the parameters. In that case

$$\boldsymbol{\theta}_t = \mathbf{u}_t \left(\{\boldsymbol{\theta}_{t-1}, \ldots, \boldsymbol{\theta}_0\}\right) = \boldsymbol{\theta}_{t-1} - \alpha(1-\beta) \sum_{t'=0}^{t-1} \beta^{t-t'-1} \nabla \mathcal{L}(\boldsymbol{\theta}_{t'}) \tag{83}$$

Similar to the case of gradient descent, the update is equivariant for any orthogonal transformation (acting at all steps)

$$\mathbf{u}_t \left(\{G\boldsymbol{\theta}_{t-1}, \ldots, G\boldsymbol{\theta}_0\}\right) = G\mathbf{u}_t \left(\{\boldsymbol{\theta}_{t-1}, \ldots, \boldsymbol{\theta}_0\}\right) \tag{84}$$

Since all groups considered in this work consists of orthogonal transformations, gradient descent (with or without momentum) is equivariant for all groups $\mathbb{G}_o, \mathbb{G}_b, \mathbb{G}_s$.

In the case of Adam optimizer, the update is equal to (we summarize time-dependent scalars in $\alpha_t$)

$$\boldsymbol{\theta}_t = \mathbf{u}_t \left(\{\boldsymbol{\theta}_{t-1}, \ldots, \boldsymbol{\theta}_0\}\right) = \boldsymbol{\theta}_{t-1} - \alpha_t \frac{\sum_{t'=0}^{t-1} \beta_1^{t-t'-1} \nabla \mathcal{L}(\boldsymbol{\theta}_{t'})}{\sqrt{\sum_{t'=0}^{t-1} \beta_2^{t-t'-1} \nabla \mathcal{L}(\boldsymbol{\theta}_{t'}) \odot \nabla \mathcal{L}(\boldsymbol{\theta}_{t'}) + \epsilon}} \tag{85}$$

Because of the denominator, this update is not equivariant for all orthogonal transformations. However, it is equivariant for signed permutations. We note that the numerator changes sign while the denominator does not, therefore the update is equivariant for sign flips. Furthermore, the update of a parameter depends only on the gradient of that parameter, therefore the update is equivariant for permutations. Therefore, Adam optimizer is equivariant for $\mathbb{G}_b, \mathbb{G}_s$, but not $\mathbb{G}_o$.

In the case of the second-order update of Equation (1), we note that Lemma 2.4 applies at $t = 0$ under assumption 2.1. Since the inverse and square root are analytic functions of positive definite matrices, then $M_0$ in Equation (1) satisfies Equation (15), provided that $\Sigma_0$ is positive definite. By induction, we can iteratively apply Theorem 2.2 to prove that the update is equivariant at all steps $t$, for all groups $\mathbb{G}_o, \mathbb{G}_b, \mathbb{G}_s$.

## G. Proof of Lemma 2.4

Given the loss $\mathcal{L}(\boldsymbol{\theta})$ and the change of variable $\boldsymbol{\theta}' = G\boldsymbol{\theta}$, we define the transformed loss $\tilde{\mathcal{L}}(\boldsymbol{\theta}) = \mathcal{L}(\boldsymbol{\theta}')$. Using the chain rule, we have

$$\nabla \tilde{\mathcal{L}}(\boldsymbol{\theta}) = G^T \nabla \mathcal{L}(\boldsymbol{\theta}') = G^T \nabla \mathcal{L}(G\boldsymbol{\theta}) \quad \implies \quad \nabla \mathcal{L}(G\boldsymbol{\theta}) = G^{T^{-1}} \nabla \tilde{\mathcal{L}}(\boldsymbol{\theta}) \tag{86}$$

by assumption, the loss is invariant, $\mathcal{L} = \tilde{\mathcal{L}}$, and $G$ is orthogonal, $G^{T^{-1}} = G$, therefore

$$\nabla \mathcal{L}(G\boldsymbol{\theta}) = G^{T^{-1}} \nabla \mathcal{L}(\boldsymbol{\theta}) = G \, \nabla \mathcal{L}(\boldsymbol{\theta}) \tag{87}$$

We note that the gradient is not equivariant in general, however it is equivariant when the transformation $G$ is orthogonal. Similarly, it follows that

$$\nabla \mathcal{L}(G\boldsymbol{\theta}) \nabla \mathcal{L}(G\boldsymbol{\theta})^T = G \, \nabla \mathcal{L}(\boldsymbol{\theta}) \nabla \mathcal{L}(\boldsymbol{\theta}) \, G^T \tag{88}$$

Using the invariance of the probability distribution, $p(G\boldsymbol{\theta}) = p(\boldsymbol{\theta})$, and a volume-preserving change of variable $\boldsymbol{\theta}' = G\boldsymbol{\theta}$, with $|\det(G)| = 1$, it is easy to show that

$$\mathbb{E}_{\boldsymbol{\theta}} \nabla \mathcal{L}(G\boldsymbol{\theta}) = \int d\boldsymbol{\theta} \, p(\boldsymbol{\theta}) \nabla \mathcal{L}(G\boldsymbol{\theta}) = \int d\boldsymbol{\theta} \, p(G\boldsymbol{\theta}) \nabla \mathcal{L}(G\boldsymbol{\theta}) = |\det(G)|^{-1} \int d\boldsymbol{\theta}' \, p(\boldsymbol{\theta}') \nabla \mathcal{L}(\boldsymbol{\theta}') = \mathbb{E}_{\boldsymbol{\theta}} \nabla \mathcal{L}(\boldsymbol{\theta}) \tag{89}$$

Using similar steps, we obtain

$$\mathbb{E}_{\boldsymbol{\theta}} \nabla \mathcal{L}(G\boldsymbol{\theta}) \nabla \mathcal{L}(G\boldsymbol{\theta})^T = \mathbb{E}_{\boldsymbol{\theta}} \nabla \mathcal{L}(\boldsymbol{\theta}) \nabla \mathcal{L}(\boldsymbol{\theta})^T \tag{90}$$

Averaging Equations (87),(88), it follows that

$$\boldsymbol{\mu} = G\boldsymbol{\mu} \quad \text{and} \quad \Sigma = G\Sigma G^T \tag{91}$$

Furthermore, since $G$ is orthogonal ($G^T G = \mathrm{I}$), any power of $\Sigma$ satisfies the same equation

$$\Sigma^m = \left(G\Sigma G^T\right)\ldots\left(G\Sigma G^T\right) = G\Sigma^m G^T \tag{92}$$

and so does any analytic function of $\Sigma$, because analytic functions are defined by power series.

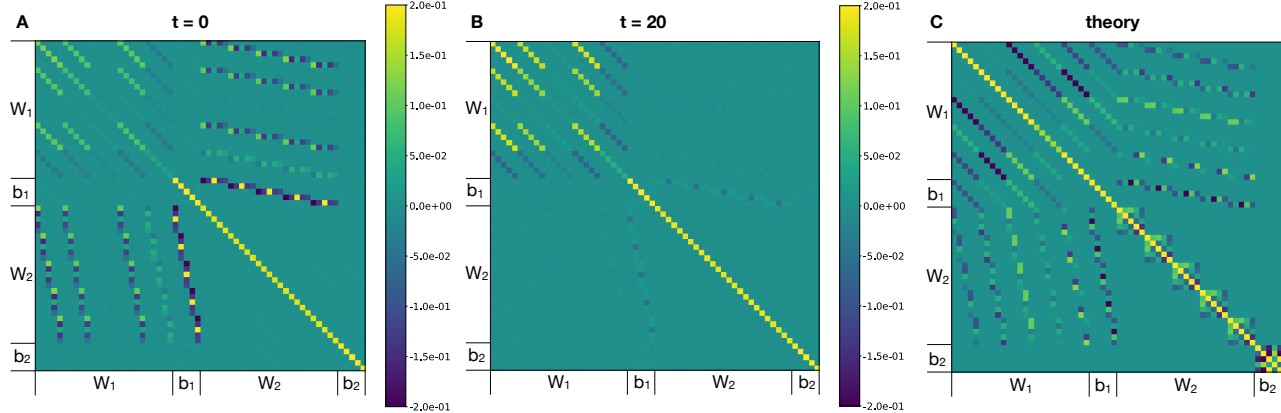

Figure 7. **Global Hessian matrix for two-layer MLP with Tanh activations**. Same as Figure 2 but shows the global Hessian instead of the gradient covariance. We highlight that theory predicts the overall structure, rather than the specific numerical values.

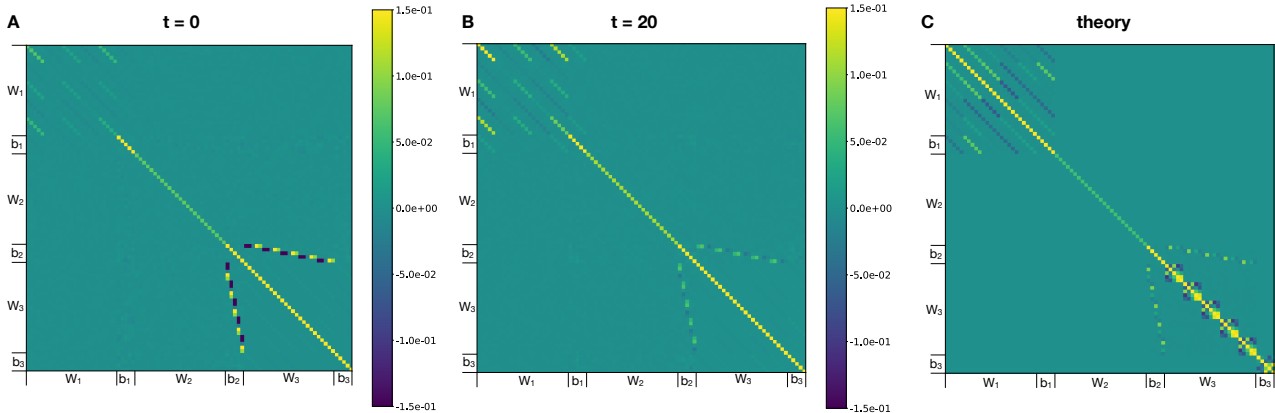

Figure 8. **Global Hessian matrix for three-layer MLP with Tanh activations**. Same as Figure 3 but shows the global Hessian instead of the gradient covariance. We highlight that theory predicts the overall structure, rather than the specific numerical values.

## H. Hessian matrices

In this section, we show that results similar to Theorem 2.4 and Figures 2, 3, 4, hold not only for the gradient covariance, but also for the Hessian. Given the loss $\mathcal{L}(\boldsymbol{\theta})$ and the change of variable $\boldsymbol{\theta}' = G\boldsymbol{\theta}$, we define the transformed loss $\tilde{\mathcal{L}}(\boldsymbol{\theta}) = \mathcal{L}(\boldsymbol{\theta}')$. Applying the chain rule twice, noting that the transformation is linear, we have

$$\nabla^2 \tilde{\mathcal{L}}(\boldsymbol{\theta}) = G^T \nabla^2 \mathcal{L}(\boldsymbol{\theta}')G = G^T \nabla^2 \mathcal{L}(G\boldsymbol{\theta})G \quad \implies \quad \nabla^2 \mathcal{L}(G\boldsymbol{\theta}) = G^{T-1}\nabla^2 \tilde{\mathcal{L}}(\boldsymbol{\theta})G^{-1} \tag{93}$$

by assumption, the loss is invariant, $\mathcal{L} = \tilde{\mathcal{L}}$, and $G$ is orthogonal, $G^{T-1} = G$, therefore

$$\nabla^2 \mathcal{L}(G\boldsymbol{\theta}) = G^{T-1}\nabla^2 \mathcal{L}(\boldsymbol{\theta})G^{-1} = G\,\nabla^2 \mathcal{L}(\boldsymbol{\theta})G^T \tag{94}$$

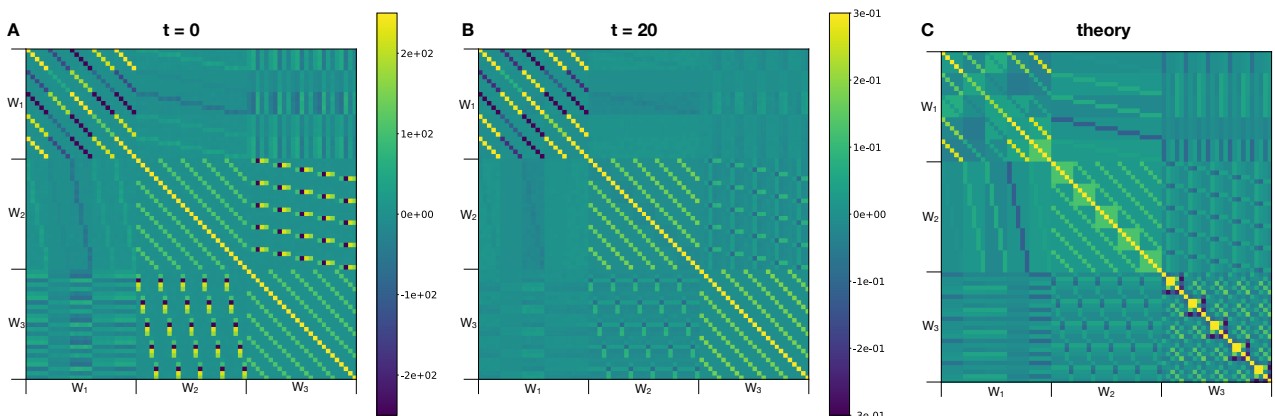

*Figure 9.* **Global Hessian matrix for three-layer MLP with ReLU activations**. Same as Figure 4 but shows the global Hessian instead of the gradient covariance. We highlight that theory predicts the overall structure, rather than the specific numerical values.

We define the averaged (or *global*) Hessian as

$$H = \mathop{\mathbb{E}}_{\boldsymbol{\theta}} \nabla^2 \mathcal{L}(\boldsymbol{\theta}) \tag{95}$$

Using steps similar to Equation (89), it is straightforward to show that the global Hessian satisfies Equations (14), (15), namely

$$H = GHG^T \quad \text{and} \quad f(H) = Gf(H)G^T \tag{96}$$

for any analytic function $f$ of the global Hessian matrix. Therefore, the statements of Lemma 2.4 for the gradient covariance hold identically for the global Hessian.

Figures 7, 8, 9 show similar results as Figures 2, 3, 4, but for the global Hessian instead of the gradient covariance. The figures show that the global Hessian has a structure similar to the gradient covariance, which is predicted equally well by our theory. Again, theory predicts the structure, rather than the specific values of the matrix. The specific values need to be estimated, as explained in Section 3.4.

## I. Proof of Theorem 3.1

We have either $V_\ell \in \mathrm{O}(d_\ell)$, the set of orthogonal matrices, or $V_\ell \in \mathrm{B}(d_\ell)$, the set of signed permutations, for $\ell = 1, \ldots, L-1$, while $V_0 = \mathrm{I}_{d_0}$ and $V_L = \mathrm{I}_{d_L}$. The gradient mean of the bias is given by. Equation (22), equal to

$$\tilde{\boldsymbol{\mu}}_\ell = V_\ell \, \tilde{\boldsymbol{\mu}}_\ell \tag{97}$$

First we note that, since $V_L = \mathrm{I}_{d_L}$, then $\tilde{\boldsymbol{\mu}}_L$ is not zero and remains undetermined. Besides that special case, this equation has a non-zero solution for $\tilde{\boldsymbol{\mu}}_\ell$ only if all transformations within the group have a common eigenvector corresponding to an eigenvalue equal to one. However, no such eigenvector exists for either groups: for orthogonal matrices, eigenvectors with eigenvalue equal to one correspond to axes of rotation and/or reflection, that can point at any direction for different orthogonal tranformations; For signed permutations, flipping simultaneously the sign of one row and the same column of a matrix translates to flipping the sign of the corresponding component in its eigenvector. Therefore signed permutations also do not have a common eigenvector and $\tilde{\boldsymbol{\mu}}_\ell$ must be equal to zero.

The mean gradient of the weights is given by Equation (21), equal to

$$\boldsymbol{\mu}_\ell = (V_{\ell-1} \otimes V_\ell) \, \boldsymbol{\mu}_\ell \tag{98}$$

This equation has a non-zero solution for $\boldsymbol{\mu}_\ell$ only if all possible products $V_{\ell-1} \otimes V_\ell$ within a group have a common eigenvector corresponding to an eigenvalue equal to one. However, for the same reasons described above, no such eigenvector exists for either groups and therefore it must be $\boldsymbol{\mu}_\ell = 0$.

Now we turn to the covariance of the gradient. The bias-bias covariance must satisfy Equation (24), equal to

$$\tilde{\tilde{\Sigma}}_{\ell\ell'} = V_\ell \, \tilde{\tilde{\Sigma}}_{\ell\ell'} \, V_{\ell'}^T \tag{99}$$

Since $\tilde{\tilde{\Sigma}}_{\ell',\ell} = \tilde{\tilde{\Sigma}}_{\ell,\ell'}^T$, we only need to study the cases $\ell \leq \ell'$. First, we note that it must be $\tilde{\tilde{\Sigma}}_{\ell,\ell'} = 0$ for all $\ell \neq \ell'$, because Equation (99) must hold for all orthogonal or signed permutation matrices $V_\ell$, $V_{\ell'}$, but that is impossible. For example, setting $V_{\ell'} = \mathrm{I}_{d_{\ell'}}$, there is no matrix $\tilde{\tilde{\Sigma}}_{\ell',\ell}$ satisfying $\tilde{\tilde{\Sigma}}_{\ell',\ell} = V_\ell \, \tilde{\tilde{\Sigma}}_{\ell',\ell}$ for all possible $V_\ell$ in the group. Next, we consider the case $\ell = \ell'$. We note that, since $V_L = \mathrm{I}_{d_L}$, then $\tilde{\tilde{\Sigma}}_{LL}$ remains undetermined. For $\ell < L$, we multiply Equation (99) by $V_\ell$ and use its orthogonality ($V_\ell^T V_\ell = \mathrm{I}_{d_\ell}$), we obtain

$$\tilde{\tilde{\Sigma}}_{\ell\ell} \, V_\ell = V_\ell \, \tilde{\tilde{\Sigma}}_{\ell\ell} \tag{100}$$

The only matrix that commutes with all orthogonal or signed permutation matrices is a scalar matrix (Stuart & Weaver, 1994), therefore we must have

$$\tilde{\tilde{\Sigma}}_{\ell\ell} = \tilde{\tilde{\phi}}_\ell \, \mathrm{I}_{d_\ell} \tag{101}$$

for some scalar value $\tilde{\tilde{\phi}}_\ell$.

The weight-weight covariance must satisfy Equation (23), equal to

$$\Sigma_{\ell\ell'} = (V_{\ell-1} \otimes V_\ell) \, \Sigma_{\ell\ell'} \, \left( V_{\ell'-1}^T \otimes V_{\ell'}^T \right) \tag{102}$$

Since $\Sigma_{\ell'\ell} = \Sigma_{\ell\ell'}^T$, we only need to study the cases $\ell \leq \ell'$. First, we study the case $\ell = \ell'$ and $\ell \neq 1, L$. In that case, using the orthogonality of $V_\ell$, $V_{\ell'}$, we rewrite Equation (102) as

$$\Sigma_{\ell\ell} \, (V_{\ell-1} \otimes V_\ell) = (V_{\ell-1} \otimes V_\ell) \, \Sigma_{\ell\ell} \tag{103}$$

Similar to the case of the bias, the only matrix that commutes with all products of orthogonal or signed permutation matrices is a scalar matrix, therefore it must be

$$\Sigma_{\ell\ell} = \phi_\ell \left( \mathrm{I}_{d_{\ell-1}} \otimes \mathrm{I}_{d_\ell} \right) \tag{104}$$

for some scalar value $\phi_\ell$. For $\ell = 1, L$, we have the following two equations

$$\Sigma_{11} = (\mathrm{I}_{d_0} \otimes V_1) \, \Sigma_{11} \left( \mathrm{I}_{d_0} \otimes V_1^T \right) \quad \text{and} \quad \Sigma_{LL} = (V_{L-1} \otimes \mathrm{I}_{d_L}) \, \Sigma_{LL} \left( V_{L-1}^T \otimes \mathrm{I}_{d_L} \right) \tag{105}$$

Using similar arguments as above, since these equations must be valid for all orthogonal or signed permutation matrices $V_1$ and $V_{L-1}$, we must have

$$\Sigma_{11} = \Phi_1 \otimes \mathrm{I}_{d_1} \quad \text{and} \quad \Sigma_{LL} = \mathrm{I}_{d_{L-1}} \otimes \Phi_L \tag{106}$$

for some matrices $\Phi_1$, $\Phi_L$. Next, we study the case $\ell' = \ell + 1$. We use the commutation matrix $K$, which is a symmetric permutation, thus $K = K^T$, $K^2 = \mathrm{I}$. The commutation matrix exchanges the factors in a Kronecker product: $K(A \otimes B)K = B \otimes A$ (see Chapter 3.7 of Magnus & Neudecker (2019)). We rewrite Equation (102) as

$$\Sigma_{\ell,\ell+1}K = (V_{\ell-1} \otimes V_\ell) \, \Sigma_{\ell,\ell+1}K \left( V_{\ell+1}^T \otimes V_\ell^T \right) \tag{107}$$

and solve for $\Sigma_{\ell,\ell+1}K$ instead of $\Sigma_{\ell,\ell+1}$. This equation must be valid for all orthogonal or signed permutation matrices $V_\ell$ ($\ell = 1, \ldots, L-1$). We show that, if either $\ell > 1$ or $\ell < L-1$, then $\Sigma_{\ell,\ell+1}K = 0 \rightarrow \Sigma_{\ell,\ell+1} = 0$. In the former case, we set $V_\ell = \mathrm{I}_{d_\ell}$, $V_{\ell+1} = \mathrm{I}_{d_{\ell+1}}$, then there is no finite matrix $\Sigma_{\ell,\ell+1}K$ that satisfies $\Sigma_{\ell,\ell+1}K = (V_{\ell-1} \otimes \mathrm{I}_{d_\ell}) \, \Sigma_{\ell,\ell+1}K$ for all $V_{\ell-1}$ in the group. In the latter case, we set $V_{\ell-1} = \mathrm{I}_{d_{\ell-1}}$ and $V_\ell = \mathrm{I}_{d_\ell}$, then there is no finite matrix $\Sigma_{\ell,\ell+1}K$ that satisfies $\Sigma_{\ell,\ell+1}K = \Sigma_{\ell,\ell+1}K \left( V_{\ell+1}^T \otimes \mathrm{I}_{d_\ell} \right)$ for all $V_{\ell+1}$ in the group. If $L = 2$, Equation (107) is equal to

$$\Sigma_{12}K = (\mathrm{I}_{d_0} \otimes V_1) \, \Sigma_{12}K \left( \mathrm{I}_{d_2} \otimes V_1^T \right) \tag{108}$$

that has the finite solution

$$\Sigma_{12}K = (\Psi_1 \otimes I_{d_1}) \quad \implies \quad \Sigma_{12} = (\Psi_1 \otimes I_{d_1}) K \tag{109}$$

for some matrix $\Psi_1$. Lastly, we study the case $\ell' > \ell + 1$. In that case, we can set $V_{\ell-1} = I_{d_{\ell-1}}$, $V_{\ell'} = I_{d_{\ell'}}$ and either $V_\ell = I_{d_\ell}$ or $V_{\ell'-1} = I_{d_{\ell'-1}}$ in Equation (102) to show that $\Sigma_{\ell,\ell'}$ must be equal to zero.

The bias-weight covariance must satisfy Equation (25), equal to

$$\tilde{\Sigma}_{\ell\ell'} = V_\ell \tilde{\Sigma}_{\ell\ell'} \left( V_{\ell'-1}^T \otimes V_{\ell'}^T \right) \tag{110}$$

We note that, in this case, $\tilde{\Sigma}_{\ell\ell'} \neq \tilde{\Sigma}_{\ell'\ell}^T$, therefore we must consider all pairs of values $\ell$, $\ell'$. For $\ell' = \ell > 1$, it must be $\tilde{\Sigma}_{\ell\ell'} = 0$. In that case, we set $V_\ell = I_{d_\ell}$ and there is no finite matrix $\tilde{\Sigma}_{\ell\ell}$ satisfying $\tilde{\Sigma}_{\ell\ell} = \tilde{\Sigma}_{\ell,\ell} \left( V_{\ell-1}^T \otimes I_{d_\ell} \right)$ for all $V_{\ell-1}$ in the group, thus $\tilde{\Sigma}_{\ell\ell} = 0$. For $\ell' = \ell + 1$ and $\ell' < L$, we set $V_\ell = I_{d_\ell}$ and there is no finite matrix $\tilde{\Sigma}_{\ell,\ell+1}$ satisfying $\tilde{\Sigma}_{\ell,\ell+1} = \tilde{\Sigma}_{\ell,\ell+1} (I_{d_\ell} \otimes V_{\ell+1})$ for all $V_{\ell+1}$ in the group, thus $\tilde{\Sigma}_{\ell,\ell+1} = 0$. For the special cases $\ell = \ell' = 1$ and $\ell' = \ell + 1 = L$, we rewrite Equation (110) as, respectively

$$\tilde{\Sigma}_{11} = V_1 \tilde{\Sigma}_{11} \left( I_{d_0} \otimes V_1^T \right) \quad \text{and} \quad \tilde{\Sigma}_{L-1,L} = V_{L-1} \tilde{\Sigma}_{L-1,L} \left( V_{L-1}^T \otimes I_{d_L} \right) \tag{111}$$

Since these equations must be valid for all orthogonal or signed permutation matrices $V_1$ and $V_{L-1}$, we must have

$$\tilde{\Sigma}_{11} = \tilde{\phi}_1^T \otimes I_{d_1} \quad \text{and} \quad \tilde{\Sigma}_{L-1,L} = I_{d_{L-1}} \otimes \tilde{\phi}_{L-1}^T \tag{112}$$

For some column vectors $\tilde{\phi}_1$ and $\tilde{\phi}_{L-1}$. For other values of $\ell$, $\ell'$, we set $V_\ell = I_{d_\ell}$ in Equation (110) and note that there is no matrix satisfying $\tilde{\Sigma}_{\ell,\ell'} = \tilde{\Sigma}_{\ell,\ell'} \left( V_{\ell'-1}^T \otimes V_{\ell'}^T \right)$ for all $V_{\ell'-1}$, $V_{\ell'}$ in the group, therefore it must be $\tilde{\Sigma}_{\ell,\ell'} = 0$.

## J. Non-symmetric activations (ReLU)

For the mean of the bias gradients, we have

$$\tilde{\mu}_\ell = \tilde{\zeta}_\ell \mathbf{1}_{d_\ell} \qquad\qquad \ell = 1, \ldots, L-1 \tag{113}$$

$$\tilde{\mu}_L = \tilde{\mathbf{z}}_L \tag{114}$$

where $\tilde{\mathbf{z}}_L \in \mathbb{R}^{d_L}$, $\tilde{\zeta}_\ell \in \mathbb{R}$ remain undetermined, $d_L + L - 1$ unknowns. For the weights

$$\boldsymbol{\mu}_1 = \mathbf{z}_0 \otimes \mathbf{1}_{d_1} = \text{Vec}\left( \mathbf{1}_{d_1} \mathbf{z}_0^T \right) \tag{115}$$

$$\boldsymbol{\mu}_\ell = \zeta_\ell (\mathbf{1}_{d_{\ell-1}} \otimes \mathbf{1}_{d_\ell}) = \zeta_\ell \text{Vec}\left( \mathbf{1}_{d_\ell} \mathbf{1}_{d_{\ell-1}}^T \right) \qquad \ell = 2, \ldots, L-1 \tag{116}$$

$$\boldsymbol{\mu}_L = \mathbf{1}_{d_{L-1}} \otimes \mathbf{z}_L = \text{Vec}\left( \mathbf{z}_L \mathbf{1}_{d_{L-1}}^T \right) \tag{117}$$

where $\mathbf{z}_0 \in \mathbb{R}^{d_0}$, $\mathbf{z}_L \in \mathbb{R}^{d_L}$, $\zeta_\ell \in \mathbb{R}$ remain undetermined, $d_0 + d_L + L - 2$ unknowns. In total, for the mean gradient we have $d_0 + 2d_L + 2L - 3$ unknowns.

For covariance of the biases, since $\tilde{\tilde{\Sigma}}_{\ell',\ell} = \tilde{\tilde{\Sigma}}_{\ell,\ell'}^T$, we only need to study the cases $\ell \leq \ell'$. We have

$$\tilde{\tilde{\Sigma}}_{\ell\ell} = \tilde{\tilde{\phi}}_\ell^{(1)} I_{d_\ell} + \frac{\tilde{\tilde{\phi}}_\ell^{(2)}}{d_\ell} \mathbf{1}_{d_\ell} \mathbf{1}_{d_\ell}^T \qquad\qquad \ell = 1, \ldots, L-1 \tag{118}$$

$$\tilde{\tilde{\Sigma}}_{\ell\ell'} = \frac{\tilde{\tilde{\omega}}_{\ell\ell'}}{\sqrt{d_\ell d_{\ell'}}} \mathbf{1}_{d_\ell} \mathbf{1}_{d_{\ell'}}^T \qquad\qquad \ell, \ell' = 1, \ldots, L-1, \quad \ell < \ell' \tag{119}$$

$$\tilde{\tilde{\Sigma}}_{\ell L} = \frac{1}{\sqrt{d_\ell}} \mathbf{1}_{d_\ell} \tilde{\tilde{\boldsymbol{\omega}}}_\ell^T \qquad\qquad \ell = 1, \ldots, L-1 \tag{120}$$

$$\tilde{\tilde{\Sigma}}_{LL} = \tilde{\tilde{\Omega}}_L \tag{121}$$

where $\tilde{\tilde{\Omega}}_{L,L}$ is a symmetric matrix of size $d_L \times d_L$, $\tilde{\tilde{\boldsymbol{\omega}}}_\ell \in \mathbb{R}^{d_L}$, $\tilde{\tilde{\omega}}_{\ell\ell'}, \tilde{\tilde{\phi}}_\ell^{(1)}, \tilde{\tilde{\phi}}_\ell^{(2)} \in \mathbb{R}$, remain undetermined, a total of $d_L(d_L + 1)/2 + (L-1)d_L + L(L-1)/2 + 2(L-1)$ unknowns.

For the weights, since $\Sigma_{\ell'\ell} = \Sigma_{\ell\ell'}^T$, we only need to study the cases $\ell \leq \ell'$. In layers $\ell = 2, \ldots, L-1$, we have

$$\Sigma_{\ell\ell} = \phi_\ell^{(1)} \left( \mathrm{I}_{d_{\ell-1}} \otimes \mathrm{I}_{d_\ell} \right) + \frac{\phi_\ell^{(2)}}{d_\ell} \left( \mathrm{I}_{d_{\ell-1}} \otimes \mathbf{1}_{d_\ell} \mathbf{1}_{d_\ell}^T \right) + \frac{\phi_\ell^{(3)}}{d_{\ell-1}} \left( \mathbf{1}_{d_{\ell-1}} \mathbf{1}_{d_{\ell-1}}^T \otimes \mathrm{I}_{d_\ell} \right) + \frac{\phi_\ell^{(4)}}{d_\ell d_{\ell-1}} \left( \mathbf{1}_{d_{\ell-1}} \mathbf{1}_{d_{\ell-1}}^T \otimes \mathbf{1}_{d_\ell} \mathbf{1}_{d_\ell}^T \right)$$
(122)

For $\ell = 2, \ldots, L-2$, we have

$$\Sigma_{\ell,\ell+1} = \frac{\psi_\ell^{(1)}}{\sqrt{d_{\ell-1} d_{\ell+1}}} \left( \mathbf{1}_{d_{\ell-1}} \mathbf{1}_{d_{\ell+1}}^T \otimes \mathrm{I}_{d_\ell} \right) K + \frac{\psi_\ell^{(2)}}{d_\ell \sqrt{d_{\ell-1} d_{\ell+1}}} \left( \mathbf{1}_{d_{\ell-1}} \mathbf{1}_{d_{\ell+1}}^T \otimes \mathbf{1}_{d_\ell} \mathbf{1}_{d_\ell}^T \right) K$$
(123)

For $2 < \ell + 1 < \ell' < L$, we have

$$\Sigma_{\ell\ell'} = \frac{\omega_{\ell,\ell'}}{\sqrt{d_{\ell-1} d_{\ell'-1} d_\ell d_{\ell'}}} \left( \mathbf{1}_{d_{\ell-1}} \mathbf{1}_{d_{\ell'-1}}^T \otimes \mathbf{1}_{d_\ell} \mathbf{1}_{d_{\ell'}}^T \right)$$
(124)

Note that this form depends on $4(L-2) + 2(L-3) + (L-3)(L-4)/2$ unknowns. Terms including the first and last layers have slightly different form

$$\Sigma_{11} = \left( \Phi_1^{(1)} \otimes \mathrm{I}_{d_1} \right) + \frac{1}{d_1} \left( \Phi_1^{(2)} \otimes \mathbf{1}_{d_1} \mathbf{1}_{d_1}^T \right)$$
(125)

$$\Sigma_{12} = \frac{1}{\sqrt{d_2}} \left( \psi_1^{(1)} \mathbf{1}_{d_2}^T \otimes \mathrm{I}_{d_1} \right) K + \frac{1}{d_1 \sqrt{d_2}} \left( \psi_1^{(2)} \mathbf{1}_{d_2}^T \otimes \mathbf{1}_{d_1} \mathbf{1}_{d_1}^T \right) K$$
(126)

$$\Sigma_{1\ell'} = \frac{1}{\sqrt{d_{\ell'-1} d_1 d_{\ell'}}} \left( \boldsymbol{\omega}_{1,\ell'} \mathbf{1}_{d_{\ell'-1}}^T \otimes \mathbf{1}_{d_1} \mathbf{1}_{d_{\ell'}}^T \right)$$
(127)

$$\Sigma_{1L} = \frac{1}{\sqrt{d_{L-1} d_1 d_L}} \left( \boldsymbol{\omega}_{1,L}^{(1)} \mathbf{1}_{d_{L-1}}^T \otimes \mathbf{1}_{d_1} \mathbf{1}_{d_L}^T \right) + \frac{1}{\sqrt{d_0 d_{L-1} d_1}} \left( \mathbf{1}_{d_0} \mathbf{1}_{d_{L-1}}^T \otimes \mathbf{1}_{d_1} \boldsymbol{\omega}_{1,L}^{(2)}{}^T \right)$$
(128)

$$\Sigma_{\ell L} = \frac{1}{\sqrt{d_{\ell-1} d_{L-1} d_\ell}} \left( \mathbf{1}_{d_{\ell-1}} \mathbf{1}_{d_{L-1}}^T \otimes \mathbf{1}_{d_\ell} \boldsymbol{\omega}_{\ell,L}^T \right)$$
(129)

$$\Sigma_{L-1,L} = \frac{1}{\sqrt{d_{L-2}}} \left( \mathbf{1}_{d_{L-2}} \psi_{L-1}^{(1)}{}^T \otimes \mathrm{I}_{d_{L-1}} \right) K + \frac{1}{d_{L-1} \sqrt{d_{L-2}}} \left( \mathbf{1}_{d_{L-2}} \psi_{L-1}^{(2)}{}^T \otimes \mathbf{1}_{d_{L-1}} \mathbf{1}_{d_{L-1}}^T \right) K$$
(130)

$$\Sigma_{LL} = \left( \mathrm{I}_{d_{L-1}} \otimes \Phi_L^{(1)} \right) + \frac{1}{d_{L-1}} \left( \mathbf{1}_{d_{L-1}} \mathbf{1}_{d_{L-1}}^T \otimes \Phi_L^{(2)} \right)$$
(131)

In the special case $L = 2$, we have instead

$$\Sigma_{12} = \left( \Psi_1^{(1)} \otimes \mathrm{I}_{d_1} \right) K + \left( \Psi_1^{(2)} \otimes \mathbf{1}_{d_1} \mathbf{1}_{d_1}^T \right) K$$
(132)

For the cross-terms, we note that $\tilde{\Sigma}_{\ell\ell'} \neq \tilde{\Sigma}_{\ell'\ell}^T$, therefore we must consider all pairs of values $\ell, \ell'$. In all layers except the first and last one, we have

$$\tilde{\Sigma}_{\ell\ell} = \frac{\tilde{\phi}_\ell^{(1)}}{\sqrt{d_{\ell-1}}} \left( \mathbf{1}_{d_{\ell-1}}^T \otimes \mathrm{I}_{d_\ell} \right) + \frac{\tilde{\phi}_\ell^{(2)}}{d_\ell \sqrt{d_{\ell-1}}} \left( \mathbf{1}_{d_{\ell-1}}^T \otimes \mathbf{1}_{d_\ell} \mathbf{1}_{d_\ell}^T \right) \qquad \ell \neq 1, L \quad (133)$$

$$\tilde{\Sigma}_{\ell,\ell+1} = \frac{\tilde{\psi}_\ell^{(1)}}{\sqrt{d_{\ell+1}}} \left( \mathrm{I}_{d_\ell} \otimes \mathbf{1}_{d_{\ell+1}}^T \right) + \frac{\tilde{\psi}_\ell^{(2)}}{d_\ell \sqrt{d_{\ell+1}}} \left( \mathbf{1}_{d_\ell} \mathbf{1}_{d_\ell}^T \otimes \mathbf{1}_{d_{\ell+1}}^T \right) \qquad \ell \neq L-1, L \quad (134)$$

$$\tilde{\Sigma}_{\ell\ell'} = \frac{\tilde{\omega}_{\ell,\ell'}}{\sqrt{d_\ell d_{\ell'-1} d_{\ell'}}} \left( \mathbf{1}_{d_\ell} \mathbf{1}_{d_{\ell'-1}}^T \otimes \mathbf{1}_{d_{\ell'}}^T \right) \qquad \ell' \neq \ell, \ell+1, \quad \ell \neq L, \quad \ell' \neq 1, L \quad (135)$$

Terms involving the first and last layer are equal to

$$\tilde{\Sigma}_{11} = \left( \tilde{\phi}_1^{(1)\,T} \otimes \mathrm{I}_{d_1} \right) + \frac{1}{d_1} \left( \tilde{\phi}_1^{(2)\,T} \otimes \mathbf{1}_{d_1} \mathbf{1}_{d_1}^T \right) \tag{136}$$

$$\tilde{\Sigma}_{\ell 1} = \frac{1}{\sqrt{d_\ell d_1}} \left( \mathbf{1}_{d_\ell} \tilde{\boldsymbol{\omega}}_{\ell 1}^T \otimes \mathbf{1}_{d_1}^T \right) \qquad\qquad \ell = 2, \ldots, L-1 \tag{137}$$

$$\tilde{\Sigma}_{L1} = \frac{1}{\sqrt{d_1}} \left( \tilde{\Omega}_{L1} \otimes \mathbf{1}_{d_1}^T \right) \tag{138}$$

$$\tilde{\Sigma}_{L\ell} = \frac{1}{\sqrt{d_{\ell-1} d_\ell}} \left( \tilde{\boldsymbol{\omega}}_{L\ell} \mathbf{1}_{d_{\ell-1}}^T \otimes \mathbf{1}_{d_\ell}^T \right) \qquad\qquad \ell = 2, \ldots, L-1 \tag{139}$$

$$\tilde{\Sigma}_{LL} = \frac{1}{\sqrt{d_{L-1}}} \left( \mathbf{1}_{d_{L-1}}^T \otimes \tilde{\Phi}_L \right) \tag{140}$$

$$\tilde{\Sigma}_{L-1,L} = \left( \mathrm{I}_{d_{L-1}} \otimes \tilde{\psi}_L^{(1)\,T} \right) + \left( \mathbf{1}_{d_{L-1}} \mathbf{1}_{d_{L-1}}^T \otimes \tilde{\psi}_L^{(2)\,T} \right) \tag{141}$$

$$\tilde{\Sigma}_{\ell L} = \frac{1}{\sqrt{d_{L-1} d_\ell}} \left( \mathbf{1}_{d_\ell} \mathbf{1}_{d_{L-1}}^T \otimes \tilde{\boldsymbol{\omega}}_{\ell L}^T \right) \qquad\qquad \ell = 2, \ldots, L-2 \tag{142}$$

**Proof**

We have $V_\ell \in \mathrm{S}(d_\ell)$, the set of permutation matrices, while $V_0 = \mathrm{I}_{d_0}$ and $V_L = \mathrm{I}_{d_L}$. We start from the gradient mean of the bias, given by Equation (22)

$$\tilde{\boldsymbol{\mu}}_\ell = V_\ell \, \tilde{\boldsymbol{\mu}}_\ell \tag{143}$$

First we note that, since $V_L = \mathrm{I}_{d_L}$, then $\tilde{\boldsymbol{\mu}}_L$ remains undetermined.

For $\ell < L$, we note that all permutations $V_\ell$ have a common eigenvector, corresponding to the eigenvalue equal to one, given by a vector with components all equal. That is the only eigenvector common to all permutations, therefore $\tilde{\boldsymbol{\mu}}_\ell = \tilde{\zeta}_\ell \mathbf{1}_{d_\ell}$ for $\ell < L$ and some scalar $\tilde{\zeta}_\ell$.

The gradient mean of the weights is given by Equation (21)

$$\boldsymbol{\mu}_\ell = (V_{\ell-1} \otimes V_\ell) \, \boldsymbol{\mu}_\ell \tag{144}$$

For $\ell \neq 1, L$, the only eigenvector common to all possible products $V_{\ell-1} \otimes V_\ell$ within the group, corresponding to an eigenvalue equal to one, is again the vector will all components equal. Therefore it must be $\boldsymbol{\mu}_\ell = \zeta_\ell (\mathbf{1}_{d_{\ell-1}} \otimes \mathbf{1}_{d_\ell})$ for some scalar $\zeta_\ell$. For $\ell = 1, L$, by similar arguments we find, respectively

$$\boldsymbol{\mu}_1 = \mathbf{z}_0 \otimes \mathbf{1}_{d_1} \quad \text{and} \quad \boldsymbol{\mu}_L = \mathbf{1}_{d_{L-1}} \otimes \mathbf{z}_L \tag{145}$$

for some column vectors $\mathbf{z}_0, \mathbf{z}_L$.

Now we turn to the covariance of the gradient. The bias-bias covariance must satisfy Equation (24), equal to

$$\tilde{\tilde{\Sigma}}_{\ell\ell'} = V_\ell \, \tilde{\tilde{\Sigma}}_{\ell\ell'} \, V_{\ell'}^T \tag{146}$$

Since $\tilde{\tilde{\Sigma}}_{\ell',\ell} = \tilde{\tilde{\Sigma}}_{\ell,\ell'}^T$, we only need to study the cases $\ell \leq \ell'$. First, we study the case $\ell = \ell'$. For $\ell = L$, we note that $V_L = \mathrm{I}_{d_L}$ and therefore $\tilde{\tilde{\Sigma}}_{LL}$ is undetermined. For $\ell < L$, we multiply Equation (146) by $V_\ell$ and use its orthogonality ($V_\ell^T V_\ell = \mathrm{I}_{d_\ell}$), we obtain

$$\tilde{\tilde{\Sigma}}_{\ell\ell} \, V_\ell = V_\ell \, \tilde{\tilde{\Sigma}}_{\ell\ell} \tag{147}$$

We note that a matrix that commutes with all permutation matrices $V_\ell$ must be the sum of a scalar matrix and a matrix with all elements equal (Stuart & Weaver, 1991). Therefore, for $\ell < L$ we have

$$\tilde{\tilde{\Sigma}}_{\ell\ell} = \tilde{\tilde{\phi}}_\ell^{(1)} \, \mathrm{I}_{d_\ell} + \frac{\tilde{\tilde{\phi}}_\ell^{(2)}}{d_\ell} \mathbf{1}_{d_\ell} \mathbf{1}_{d_\ell}^T \tag{148}$$

for some scalars $\tilde{\tilde{\phi}}_\ell^{(1)}, \tilde{\tilde{\phi}}_\ell^{(2)}$. For $\ell \neq \ell'$, since the only eigenvector common to all permutations is a vector with all components equal, then Equation (146) is satisfied by

$$\tilde{\tilde{\Sigma}}_{\ell\ell'} = \frac{\tilde{\tilde{\omega}}_{\ell\ell'}}{\sqrt{d_\ell d_{\ell'}}} \mathbf{1}_{d_\ell} \mathbf{1}_{d_{\ell'}}^T, \quad \text{if } \ell < \ell' < L, \text{ and} \quad \tilde{\tilde{\Sigma}}_{\ell L} = \frac{1}{\sqrt{d_\ell}} \mathbf{1}_{d_\ell} \tilde{\tilde{\omega}}_\ell^T \quad \text{if } \ell < L \tag{149}$$

For some scalar $\tilde{\tilde{\omega}}_{\ell\ell'}$ and column vector $\tilde{\tilde{\omega}}_\ell$.

The weight-weight covariance must satisfy Equation (23), equal to

$$\Sigma_{\ell\ell'} = (V_{\ell-1} \otimes V_\ell) \Sigma_{\ell\ell'} \left( V_{\ell'-1}^T \otimes V_{\ell'}^T \right) \tag{150}$$

Since $\Sigma_{\ell'\ell} = \Sigma_{\ell\ell'}^T$, we only need to study the cases $\ell \leq \ell'$. First, we study the case $\ell = \ell'$ and $\ell \neq 1, L$. In that case, using the orthogonality of $V_\ell, V_{\ell'}$, we rewrite Equation (150) as

$$\Sigma_{\ell\ell} (V_{\ell-1} \otimes V_\ell) = (V_{\ell-1} \otimes V_\ell) \Sigma_{\ell\ell} \tag{151}$$

Similar to the case of the bias, a matrix that commutes with a product of permutation matrices must have the form (Stuart & Weaver, 1991)

$$\Sigma_{\ell\ell} = \phi_\ell^{(1)} \left( I_{d_{\ell-1}} \otimes I_{d_\ell} \right) + \frac{\phi_\ell^{(2)}}{d_\ell} \left( I_{d_{\ell-1}} \otimes \mathbf{1}_{d_\ell} \mathbf{1}_{d_\ell}^T \right) + \frac{\phi_\ell^{(3)}}{d_{\ell-1}} \left( \mathbf{1}_{d_{\ell-1}} \mathbf{1}_{d_{\ell-1}}^T \otimes I_{d_\ell} \right) + \frac{\phi_\ell^{(4)}}{d_\ell d_{\ell-1}} \left( \mathbf{1}_{d_{\ell-1}} \mathbf{1}_{d_{\ell-1}}^T \otimes \mathbf{1}_{d_\ell} \mathbf{1}_{d_\ell}^T \right) \tag{152}$$

For some scalars $\phi_\ell^{(1)}, \phi_\ell^{(2)}, \phi_\ell^{(3)}, \phi_\ell^{(4)}$. For $\ell = 1, L$, since $V_0 = I_{d_0}$, $V_{L-1} = I_{d_{L-1}}$, the covariance block satisfies, respectively

$$\Sigma_{11} = (I_{d_0} \otimes V_1) \Sigma_{11} (I_{d_0} \otimes V_1^T) \quad \text{and} \quad \Sigma_{LL} = (V_{L-1} \otimes I_{d_L}) \Sigma_{LL} (V_{L-1}^T \otimes I_{d_L}) \tag{153}$$

Using similar arguments as above, since these equations must be valid for all permutation matrices $V_1$ and $V_{L-1}$, we must have

$$\Sigma_{11} = \left( \Phi_1^{(1)} \otimes I_{d_1} \right) + \frac{1}{d_1} \left( \Phi_1^{(2)} \otimes \mathbf{1}_{d_1} \mathbf{1}_{d_1}^T \right) \quad \text{and} \quad \Sigma_{LL} = \left( I_{d_{L-1}} \otimes \Phi_L^{(1)} \right) + \frac{1}{d_{L-1}} \left( \mathbf{1}_{d_{L-1}} \mathbf{1}_{d_{L-1}}^T \otimes \Phi_L^{(2)} \right) \tag{154}$$

for some matrices $\Phi_1^{(1)}, \Phi_1^{(2)}, \Phi_L^{(1)}, \Phi_L^{(2)}$. Next, we study the case $\ell' = \ell + 1$. We use the commutation matrix $K$, which is a symmetric permutation, thus $K = K^T$, $K^2 = I$. The commutation matrix exchanges the factors in a Kronecker product: $K(A \otimes B)K = B \otimes A$ (see Chapter 3.7 of Magnus & Neudecker (2019)). We rewrite Equation (150) as

$$\Sigma_{\ell,\ell+1} K = (V_{\ell-1} \otimes V_\ell) \Sigma_{\ell,\ell+1} K \left( V_{\ell+1}^T \otimes V_\ell^T \right) \tag{155}$$

and solve for $\Sigma_{\ell,\ell+1} K$ instead of $\Sigma_{\ell,\ell+1}$. We start with $\ell \neq 1, L-1$, then this equation must be valid for all permutation matrices $V_{\ell-1}, V_\ell, V_{\ell+1}$. Using similar arguments as above, we find

$$\Sigma_{\ell,\ell+1} = \frac{\psi_\ell^{(1)}}{\sqrt{d_{\ell-1} d_{\ell+1}}} \left( \mathbf{1}_{d_{\ell-1}} \mathbf{1}_{d_{\ell+1}}^T \otimes I_{d_\ell} \right) K + \frac{\psi_\ell^{(2)}}{d_\ell \sqrt{d_{\ell-1} d_{\ell+1}}} \left( \mathbf{1}_{d_{\ell-1}} \mathbf{1}_{d_{\ell+1}}^T \otimes \mathbf{1}_{d_\ell} \mathbf{1}_{d_\ell}^T \right) K \tag{156}$$

for some scalars $\psi_\ell^{(1)}, \psi_\ell^{(2)}$. For $\ell = 1, L-1$, Equation (155) becomes, respectively

$$\Sigma_{12} K = (I_{d_0} \otimes V_1) \Sigma_{12} K \left( V_2^T \otimes V_1^T \right) \quad \text{and} \quad \Sigma_{L-1,L} K = (V_{L-2} \otimes V_{L-1}) \Sigma_{L-1,L} K \left( I_{d_L} \otimes V_{L-1}^T \right) \tag{157}$$

These equations must be valid for all permutations $V_1, V_2$ and $V_{L-2}, V_{L-1}$. We find

$$\Sigma_{12} = \frac{1}{\sqrt{d_2}} \left( \psi_1^{(1)} \mathbf{1}_{d_2}^T \otimes I_{d_1} \right) K + \frac{1}{d_1 \sqrt{d_2}} \left( \psi_1^{(2)} \mathbf{1}_{d_2}^T \otimes \mathbf{1}_{d_1} \mathbf{1}_{d_1}^T \right) K \tag{158}$$

$$\Sigma_{L-1,L} = \frac{1}{\sqrt{d_{L-2}}} \left( \mathbf{1}_{d_{L-2}} \psi_{L-1}^{(1)}{}^T \otimes I_{d_{L-1}} \right) K + \frac{1}{d_{L-1} \sqrt{d_{L-2}}} \left( \mathbf{1}_{d_{L-2}} \psi_{L-1}^{(2)}{}^T \otimes \mathbf{1}_{d_{L-1}} \mathbf{1}_{d_{L-1}}^T \right) K \tag{159}$$

For some column vectors $\psi_1^{(1)}, \psi_1^{(2)}, \psi_{L-1}^{(1)}, \psi_{L-1}^2$. The case $L = 2$ is distinct, since Equation (155) is equal to

$$\Sigma_{12} K = (I_{d_0} \otimes V_1) \, \Sigma_{12} K \left( I_{d_2} \otimes V_1^T \right) \tag{160}$$

that has the solution

$$\Sigma_{12} = \left( \Psi_1^{(1)} \otimes I_{d_1} \right) K + \left( \Psi_1^{(2)} \otimes \mathbf{1}_{d_1} \mathbf{1}_{d_1}^T \right) K \tag{161}$$

for some matrix $\Psi_1$. Lastly, we study the case $\ell' > \ell + 1$, starting with $\ell > 1$ and $\ell' < L$. In that case, Equation (150) must be valid for all different possible permutations $V_{\ell-1}, V_\ell, V_{\ell'-1}, V_{\ell'}$. The solution is

$$\Sigma_{\ell\ell'} = \frac{\omega_{\ell,\ell'}}{\sqrt{d_{\ell-1} d_{\ell'-1} d_\ell d_{\ell'}}} \left( \mathbf{1}_{d_{\ell-1}} \mathbf{1}_{d_{\ell'-1}}^T \otimes \mathbf{1}_{d_\ell} \mathbf{1}_{d_{\ell'}}^T \right) \tag{162}$$

If $\ell = 1$, $\ell' < L$, Equation (150) must be valid for all different possible permutations $V_\ell, V_{\ell'-1}, V_{\ell'}$, and

$$\Sigma_{1\ell'} = \frac{1}{\sqrt{d_{\ell'-1} d_1 d_{\ell'}}} \left( \omega_{1,\ell'} \mathbf{1}_{d_{\ell'-1}}^T \otimes \mathbf{1}_{d_1} \mathbf{1}_{d_{\ell'}}^T \right) \tag{163}$$

If $\ell > 1$, $\ell' = L$, Equation (150) must be valid for all different possible permutations $V_{\ell-1}, V_\ell, V_{L-1}$, and

$$\Sigma_{\ell L} = \frac{1}{\sqrt{d_{\ell-1} d_{L-1} d_\ell}} \left( \mathbf{1}_{d_{\ell-1}} \mathbf{1}_{d_{L-1}}^T \otimes \mathbf{1}_{d_\ell} \omega_{\ell,L}^T \right) \tag{164}$$

Lastly, for $\ell = 1$, $\ell' = L$ (with $L > 2$), Equation (150) must be valid for all different possible permutations $V_1, V_{L-1}$, and

$$\Sigma_{1L} = \frac{1}{\sqrt{d_{L-1} d_1 d_L}} \left( \omega_{1,L}^{(1)} \mathbf{1}_{d_{L-1}}^T \otimes \mathbf{1}_{d_1} \mathbf{1}_{d_L}^T \right) + \frac{1}{\sqrt{d_0 d_{L-1} d_1}} \left( \mathbf{1}_{d_0} \mathbf{1}_{d_{L-1}}^T \otimes \mathbf{1}_{d_1} \omega_{1,L}^{(2)}{}^T \right) \tag{165}$$

The bias-weight covariance must satisfy Equation (25), equal to

$$\tilde{\Sigma}_{\ell\ell'} = V_\ell \, \tilde{\Sigma}_{\ell\ell'} \left( V_{\ell'-1}^T \otimes V_{\ell'}^T \right) \tag{166}$$

We note that, in this case, $\tilde{\Sigma}_{\ell\ell'} \neq \tilde{\Sigma}_{\ell'\ell}^T$, therefore we must consider all pairs of values $\ell, \ell'$. We start with $\ell' = \ell$. For $\ell \neq 1, L$, Equation (166) must hold for all possible different permutation matrices $V_{\ell-1}, V_\ell$, therefore

$$\tilde{\Sigma}_{\ell\ell} = \frac{\tilde{\phi}_\ell^{(1)}}{\sqrt{d_{\ell-1}}} \left( \mathbf{1}_{d_{\ell-1}}^T \otimes I_{d_\ell} \right) + \frac{\tilde{\phi}_\ell^{(2)}}{d_\ell \sqrt{d_{\ell-1}}} \left( \mathbf{1}_{d_{\ell-1}}^T \otimes \mathbf{1}_{d_\ell} \mathbf{1}_{d_\ell}^T \right) \tag{167}$$

for some scalars $\tilde{\phi}_\ell^{(1)}, \tilde{\phi}_\ell^{(2)}$. For the special cases $\ell = \ell' = 1$ and $\ell' = \ell = L$, we rewrite Equation (166) as, respectively

$$\tilde{\Sigma}_{11} = V_1 \, \tilde{\Sigma}_{11} \left( I_{d_0} \otimes V_1^T \right) \quad \text{and} \quad \tilde{\Sigma}_{LL} = \tilde{\Sigma}_{LL} \left( V_{L-1}^T \otimes I_{d_L} \right) \tag{168}$$

Since these equations must be valid for all permutation matrices $V_1$ and $V_{L-1}$, we must have

$$\tilde{\Sigma}_{11} = \left( \tilde{\phi}_1^{(1)}{}^T \otimes I_{d_1} \right) + \frac{1}{d_1} \left( \tilde{\phi}_1^{(2)}{}^T \otimes \mathbf{1}_{d_1} \mathbf{1}_{d_1}^T \right) \quad \text{and} \quad \tilde{\Sigma}_{LL} = \frac{1}{\sqrt{d_{L-1}}} \left( \mathbf{1}_{d_{L-1}}^T \otimes \tilde{\Phi}_L \right) \tag{169}$$

For some column vectors $\tilde{\phi}_1^{(1)}, \tilde{\phi}_1^{(2)}$ and some matrix $\tilde{\Phi}_L$. Next, we look at the case $\ell' = \ell + 1$. For $\ell < L - 1$, we rewrite Equation (166) as

$$\tilde{\Sigma}_{\ell,\ell+1} = V_\ell \, \tilde{\Sigma}_{\ell,\ell+1} \left( V_\ell^T \otimes V_{\ell+1}^T \right) \tag{170}$$

This equation must be valid for all permutations $V_\ell, V_{\ell+1}$, therefore

$$\tilde{\Sigma}_{\ell,\ell+1} = \frac{\tilde{\psi}_\ell^{(1)}}{\sqrt{d_{\ell+1}}} \left( I_{d_\ell} \otimes \mathbf{1}_{d_{\ell+1}}^T \right) + \frac{\tilde{\psi}_\ell^{(2)}}{d_\ell \sqrt{d_{\ell+1}}} \left( \mathbf{1}_{d_\ell} \mathbf{1}_{d_\ell}^T \otimes \mathbf{1}_{d_{\ell+1}}^T \right) \tag{171}$$

for some scalars $\tilde{\psi}_\ell^{(1)}$, $\tilde{\psi}_\ell^{(2)}$. For the special case $\ell' = \ell + 1 = L$, we rewrite Equation (166) as

$$\tilde{\Sigma}_{L-1,L} = V_{L-1} \, \tilde{\Sigma}_{L-1,L} \left( V_{L-1}^T \otimes \mathrm{I}_{d_L} \right) \tag{172}$$

This equation must be valid for all permutations $V_{L-1}$, therefore we find

$$\tilde{\Sigma}_{L-1,L} = \left( \mathrm{I}_{d_{L-1}} \otimes \tilde{\psi}_L^{(1)\,T} \right) + \left( \mathbf{1}_{d_{L-1}} \mathbf{1}_{d_{L-1}}^T \otimes \tilde{\psi}_L^{(2)\,T} \right) \tag{173}$$

for some column vectors $\tilde{\psi}_L^{(1)}$, $\tilde{\psi}_L^{(2)}$. Lastly, we study the case $\ell' \neq \ell, \ell + 1$. For $\ell \neq L$ and $\ell' \neq 1, L$, Equation (166) must hold for all possible different permutation matrices $V_\ell$, $V_{\ell'-1}$, $V_{\ell'}$, therefore we have

$$\tilde{\Sigma}_{\ell\ell'} = \frac{\tilde{\omega}_{\ell,\ell'}}{\sqrt{d_\ell d_{\ell'-1} d_{\ell'}}} \left( \mathbf{1}_{d_\ell} \mathbf{1}_{d_{\ell'-1}}^T \otimes \mathbf{1}_{d_{\ell'}}^T \right) \tag{174}$$

For some scalar $\tilde{\omega}_{\ell,\ell'}$. For $\ell = L$ and $\ell' \neq 1, L$, Equation (166) becomes

$$\tilde{\Sigma}_{L\ell'} = \tilde{\Sigma}_{L\ell'} \left( V_{\ell'-1}^T \otimes V_{\ell'}^T \right) \tag{175}$$

This equation must hold for all possible different permutation matrices $V_{\ell'-1}$, $V_{\ell'}$, therefore we have

$$\tilde{\Sigma}_{L\ell'} = \frac{1}{\sqrt{d_{\ell'-1} d_\ell'}} \left( \tilde{\omega}_{L\ell'} \mathbf{1}_{d_{\ell'-1}}^T \otimes \mathbf{1}_{d_\ell'}^T \right) \tag{176}$$

For some column vector $\tilde{\omega}_{L\ell'}$. For $\ell \neq 1, L$ and $\ell' = 1$, Equation (166) becomes

$$\tilde{\Sigma}_{\ell 1} = V_\ell \, \tilde{\Sigma}_{\ell 1} \left( \mathrm{I}_{d_0} \otimes V_1^T \right) \tag{177}$$

This equation must hold for all possible different permutation matrices $V_1$, $V_\ell$, therefore we have

$$\tilde{\Sigma}_{\ell 1} = \frac{1}{\sqrt{d_\ell d_1}} \left( \mathbf{1}_{d_\ell} \tilde{\omega}_{\ell 1}^T \otimes \mathbf{1}_{d_1}^T \right) \tag{178}$$

For some column vector $\tilde{\omega}_{\ell 1}$. For $\ell \neq 1, L-1, L$ and $\ell' = L$, Equation (166) becomes

$$\tilde{\Sigma}_{\ell L} = V_\ell \, \tilde{\Sigma}_{\ell L} \left( V_{L-1}^T \otimes \mathrm{I}_{d_L} \right) \tag{179}$$

This equation must hold for all possible different permutation matrices $V_\ell$, $V_{L-1}$, therefore we have

$$\tilde{\Sigma}_{\ell L} = \frac{1}{\sqrt{d_{L-1} d_\ell}} \left( \mathbf{1}_{d_\ell} \mathbf{1}_{d_{L-1}}^T \otimes \tilde{\omega}_{\ell L}^T \right) \tag{180}$$

For some column vector $\tilde{\omega}_{\ell L}$. Lastly, we have the case $\ell = L$, $\ell' = 1$. In that case Equation (166) becomes

$$\tilde{\Sigma}_{L1} = \tilde{\Sigma}_{L1} \left( \mathrm{I}_{d_0} \otimes V_1^T \right) \tag{181}$$

This equation must hold for all possible different permutation matrices $V_1$, therefore we have

$$\tilde{\Sigma}_{L1} = \frac{1}{\sqrt{d_1}} \left( \tilde{\Omega}_{L1} \otimes \mathbf{1}_{d_1}^T \right) \tag{182}$$

for some matrix $\tilde{\Omega}_{L1}$.

## K. Estimation of factors

In this section, we provide some details on how to estimate the factors of the gradient covariance. First, we provide a derivation of Equations (37),(38),(39) in the main text. We then describe how to improve the estimates by adding momentum, and provide an empirical analysis of the estimation error (Figure 10).

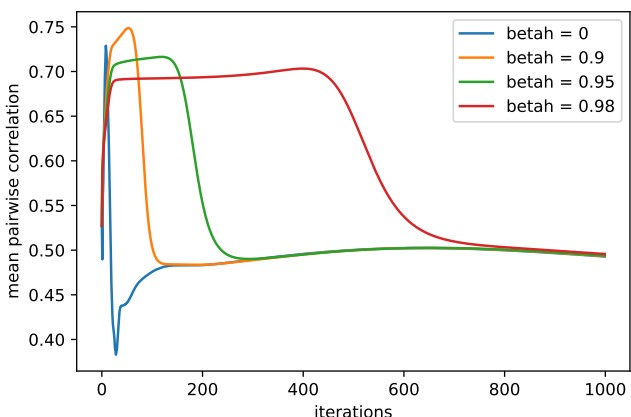

*Figure 10.* **The factors of the covariance can be estimated by a single model.** Mean pairwise correlation of estimated factors across 10 independently initialized models (45 pairs of models) as a function of training iteration (gradient descent), for different values of the momentum parameter $\beta_h$. A correlation value close to 1 would imply that hyperparameter estimates agree across independent models. Without momentum ($\beta_h = 0$), the hypeparameters of independent models display a correlation of 0.5. With momentum, correlations reach about 0.7.

We consider the example of a two-layer MLP with Tanh activations and no bias, but a similar procedure can be followed for other cases. The covariance is described in Theorem 3.1, in this case is equal to

$$\Sigma = \begin{pmatrix} \Sigma_{11} & \Sigma_{12} \\ \Sigma_{12}^T & \Sigma_{22} \end{pmatrix} = \begin{pmatrix} \Phi_1 \otimes I_{d_1} & (\Psi_1 \otimes I_{d_1}) K \\ K (\Psi_1^T \otimes I_{d_1}) & I_{d_1} \otimes \Phi_2 \end{pmatrix} \tag{183}$$

First, we assume that the covariance $\Sigma$ is given, and we are asked to obtain the factors $\{\Phi_1, \Phi_2, \Psi_1\}$. Later we consider the case in which we have an estimate of $\Sigma$, instead of its true value. Given $\Sigma$, to obtain the factors $\{\Phi_1, \Phi_2, \Psi_1\}$ we need to invert the following expressions, describing each block of the covariance

$$\Sigma_{11} = \Phi_1 \otimes I_{d_1} \qquad \Sigma_{22} = I_{d_1} \otimes \Phi_2 \qquad \Sigma_{12} = (\Psi_1 \otimes I_{d_1}) K \tag{184}$$

It is convenient to rewrite these equations including the indices of all matrix elements

$$(\Sigma_{11})_{ij,kl} = (\Phi_1)_{jl}\, \delta_{ik} \qquad (\Sigma_{22})_{ij,kl} = \delta_{jl}\, (\Phi_2)_{ik} \qquad (\Sigma_{12})_{ij,kl} = (\Psi_1)_{jk}\, \delta_{il} \tag{185}$$

The problem of inverting these equations is underparameterized, therefore we look for the least square solution. That is equal to

$$(\Phi_1)_{jl} = \frac{1}{d_1} \sum_{i=1}^{d_1} (\Sigma_{11})_{ij,il} \tag{186}$$

$$(\Phi_2)_{ik} = \frac{1}{d_1} \sum_{j=1}^{d_1} (\Sigma_{22})_{ij,kj} \tag{187}$$

$$(\Psi_1)_{jk} = \frac{1}{d_1} \sum_{i=1}^{d_1} (\Sigma_{12})_{ij,ki} \tag{188}$$

In practice, the covariance $\Sigma$ is not given, we only have access to an estimate from the observation of a gradient vector. We note that the mean gradient is zero in this case (Theorem 3.1), therefore, we substitute the values of $\Sigma_{11}, \Sigma_{12}, \Sigma_{22}$ with the observed outer product of the observed gradient, and we obtain Equations (37),(38),(39) in the main text.

These equations show sums over neurons of the first layer, of dimension $d_1$. The larger is $d_1$, the lower is the expected error in the estimates, because there will be a correspondingly large number of repeated values in the gradient covariance, therefore the value can be estimated by summing over the corresponding pairs of parameters. In practice, the expected value of the error in the estimate depends on the distribution of the gradients, which is unknown and likely non-Gaussian, therefore it is not easy to estimate.

To improve the estimate of the factors, we use momentum, a standard method for noise reduction. Equations (37),(38),(39) provide an estimate of the factors for a fixed iteration step during training. We rewrite those equations here in matrix form for convenience

$$\Phi_1 = \frac{1}{d_1} \left( \frac{\partial \mathcal{L}}{\partial W_1} \right)^T \left( \frac{\partial \mathcal{L}}{\partial W_1} \right) \tag{189}$$

$$\Phi_2 = \frac{1}{d_1} \left( \frac{\partial \mathcal{L}}{\partial W_2} \right) \left( \frac{\partial \mathcal{L}}{\partial W_2} \right)^T \tag{190}$$

$$\Psi_1 = \frac{1}{d_1} \left( \frac{\partial \mathcal{L}}{\partial W_1} \right)^T \left( \frac{\partial \mathcal{L}}{\partial W_2} \right)^T \tag{191}$$

Across training iterations, instead of using the equations above, we use the following standard instance of momentum for estimating the factors at step $t$ given the estimate at step $t-1$

$$(\Phi_1)_t = \beta_h (\Phi_1)_{t-1} + \frac{1-\beta_h}{d_1} \left( \frac{\partial \mathcal{L}}{\partial W_1} \right)_t^T \left( \frac{\partial \mathcal{L}}{\partial W_1} \right)_t \tag{192}$$

$$(\Phi_2)_t = \beta_h (\Phi_2)_{t-1} + \frac{1-\beta_h}{d_1} \left( \frac{\partial \mathcal{L}}{\partial W_2} \right)_t \left( \frac{\partial \mathcal{L}}{\partial W_2} \right)_t^T \tag{193}$$

$$(\Psi_1)_t = \beta_h (\Psi_1)_{t-1} + \frac{1-\beta_h}{d_1} \left( \frac{\partial \mathcal{L}}{\partial W_1} \right)^T \left( \frac{\partial \mathcal{L}}{\partial W_2} \right)^T \tag{194}$$

All factors are set to zero at $t = 0$, and estimates at step $t$ are divided by the factor $(1 - \beta_h^t)$ to remove the bias in the estimate, as in standard practice (Kingma & Ba, 2014).

We study the estimation error by running 10 models, each one with its own random initialization and its own individual esimate of the hyperparametsrs. We training each model separately by gradient descent. At each iteration, we estimate the hyperparamaters of each model and save it into a flattened vector containing all factors. Then, we compare the models by computing correlation between each pair of models (a total of 45 pairs) and compute the mean correlation across pairs. A large correlation would imply that estimates of different models agree, suggesting that the estimates are good. We run the same model described in Section 4, with layer widths $d_0 = 100$, $d_1 = 70$, $d_2 = 40$, for a total of 9800 parameters, 5000 training input data points drawn from a Gaussian distribution with badly conditioned covariance and the output given by a teacher network with identical architecture and random weights (see Section 4 for details).

Figure 10 shows the mean correlation as a function of training iterations for different values of $\beta_h$. Without momentum ($\beta_h = 0$), the factors estimated from different individual models display a correlation of about 0.5, showing that individual models estimate similar but not identical factors. The introduction of momentum allows increasing the correlation up to about 0.7, therefore it provides a significant improvement. However, the correlation drops back to about 0.5 after a number of iterations that depends on the value of $\beta_h$. We note that the number of steps it takes to drop is equal to about 10 times the intrinsic timescale of momentum, equal to $(1 - \beta_h)^{-1}$.

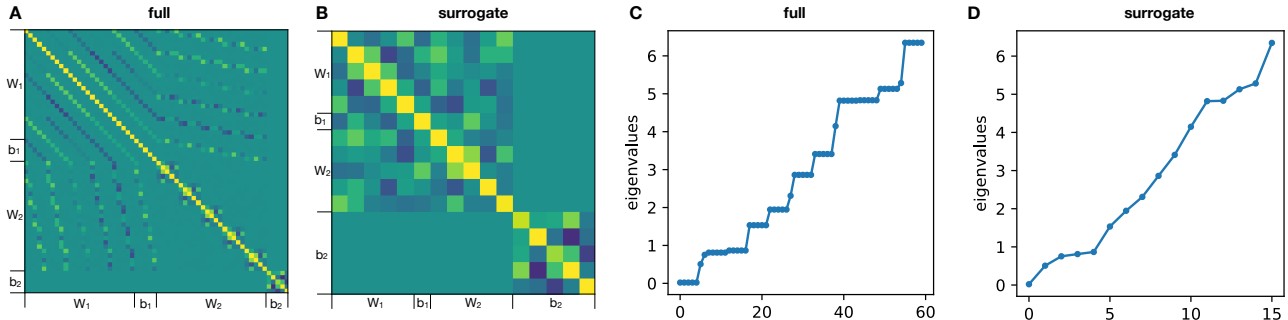

*Figure 11.* Comparison of full (A) and surrogate (B) matrix for a two-layer MLP with Tanh activations. Panels C and D compare the eigenvalues of the two matrices. We note that the eigenvalues are identical, although some of the eigenvalues of the full matrix are repeated multiple times.

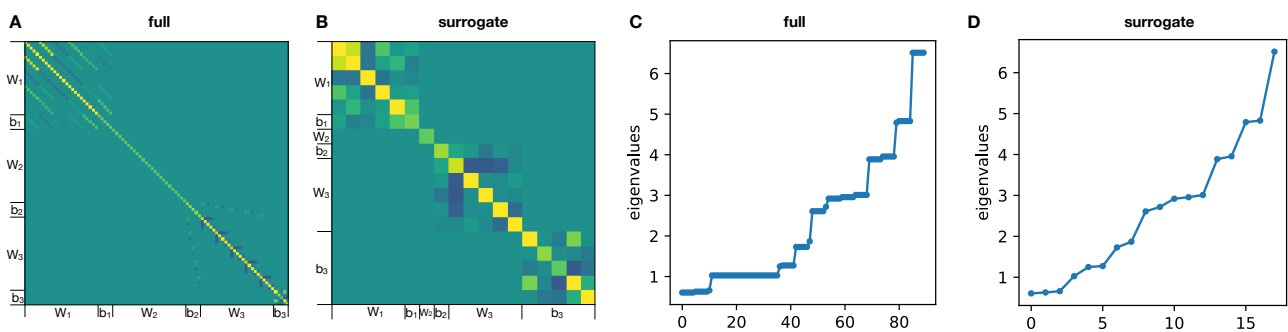

*Figure 12.* Similar to Figure 11, for a three-layer MLP with Tanh activations.

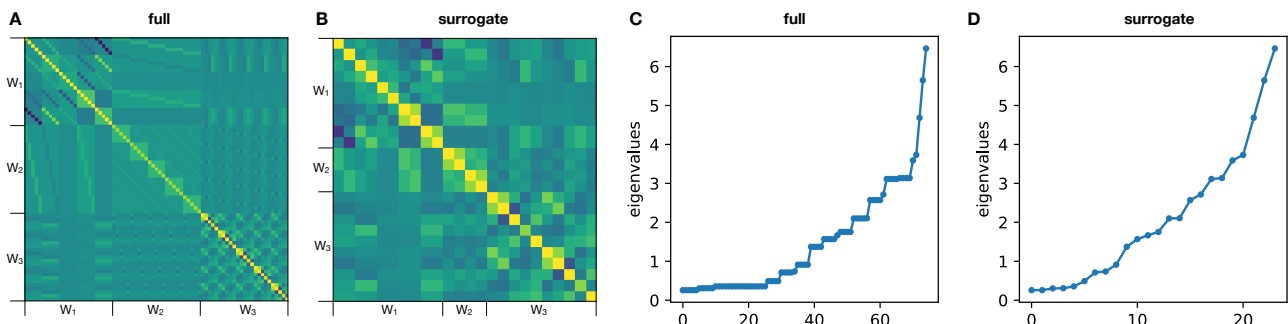

*Figure 13.* Similar to Figures 11 and 12, for a three-layer MLP with ReLU activations.

## L. Surrogate matrix

In this section, we describe how to perform operations on the covariance matrix using a smaller *surrogate* matrix. In particular, we need to compute its inverse square root, that is required by the second-order update of Equation (1). First we recall that, using either Theorem 3.1 (Tanh activation) or Theorem 3.2 (ReLU activation), we can describe the covariance matrix by a set of factors. We consider the case of Tanh, but a similar argument can be used for the case of ReLU. We define the flattened vector of all factors by

$$\phi = \text{Vec}\left(\{\phi_2, \ldots, \phi_{L-1}\}, \{\tilde{\tilde{\phi}}_1, \ldots, \tilde{\tilde{\phi}}_{L-1}\}, \tilde{\phi}_1, \tilde{\phi}_{L-1}, \Phi_1, \Phi_L, \tilde{\tilde{\Phi}}_L, \Psi_1\right) \tag{195}$$

Using Theorem 3.1, We can compute the covariance matrix given the factors $\phi$ and the layer widths $\{d_\ell\}$, and *vice versa*.

$$\phi, \{d_1, \ldots, d_{L-1}\} \longleftrightarrow \Sigma \tag{196}$$

The next important observation is that, by Lemma 2.4, Equation (15), any analytic function of the covariance matrix must have the same structure of the covariance itself. Therefore, provided that the covariance is positive definite, the inverse square root is also described by a vector of factors, identified by the "isr" superscript

$$\phi^{\text{isr}} = \text{Vec}\left(\{\phi_2^{\text{isr}}, \ldots, \phi_{L-1}^{\text{isr}}\}, \{\tilde{\tilde{\phi}}_1^{\text{isr}}, \ldots, \tilde{\tilde{\phi}}_{L-1}^{\text{isr}}\}, \tilde{\phi}_1^{\text{isr}}, \tilde{\phi}_{L-1}^{\text{isr}}, \Phi_1^{\text{isr}}, \Phi_L^{\text{isr}}, \tilde{\tilde{\Phi}}_L^{\text{isr}}, \Psi_1^{\text{isr}}\right) \tag{197}$$

Also in this case, we can compute the matrix given the factors and the layer widths, and *vice versa*

$$\phi^{\text{isr}}, \{d_1, \ldots, d_{L-1}\} \longleftrightarrow \Sigma^{-\frac{1}{2}} \tag{198}$$

A naive approach for computing the factors $\phi^{\text{isr}}$ of the inverse square root $\Sigma^{-\frac{1}{2}}$ would be to compute $\Sigma$ from $\phi$, then its inverse square root $\Sigma^{-\frac{1}{2}}$ and finally $\phi^{\text{isr}}$ from $\Sigma^{-\frac{1}{2}}$.

$$\phi, \{d_1, \ldots, d_{L-1}\} \longrightarrow \Sigma \longrightarrow \Sigma^{-\frac{1}{2}} \longrightarrow \phi^{\text{isr}} \tag{199}$$

However, the matrix $\Sigma$ may be very large and exceed the available memory. Furthermore, computing its inverse square root may be computationally very expensive. Therefore, we would like to compute $\phi^{\text{isr}}$ directly from $\phi$, without ever computing and square rooting / inverting the big matrix $\Sigma$.

We solve this problem by noting that the factors do not depend on the choice of the layer widths. In other words, any alternative choice $\{d_\ell^*\}$ of the layer widths would give the same outcome $\phi^{\text{isr}}$.

$$\phi, \{d_1^*, \ldots, d_{L-1}^*\} \longrightarrow \Lambda \longrightarrow \Lambda^{-\frac{1}{2}} \longrightarrow \phi^{\text{isr}} \tag{200}$$

where $\Lambda$ is a *surrogate* matrix with the same factors as $\Sigma$ but different layer widths. Therefore, we can set the layer widths to very small numbers, corresponding to a surrogate matrix $\Lambda$ that is easy to store in memory and to square root / invert. In particular, the smallest number that retains the factors is $d_\ell^* = 1$ for case of Tanh and $d_\ell^* = 2$ for the case of ReLU.

For example, in the case of a two-layer MLP ($L = 2$) with Tanh activation and no bias, the full matrix has size $(d_0 d_1 + d_1 d_2) \times (d_0 d_1 + d_1 d_2)$ and is equal to

$$\Sigma = \begin{pmatrix} \Phi_1 \otimes I_{d_1} & (\Psi_1 \otimes I_{d_1}) K \\ K (\Psi_1^T \otimes I_{d_1}) & I_{d_1} \otimes \Phi_2 \end{pmatrix} \tag{201}$$

while the surrogate matrix has size $(d_0 + d_2) \times (d_0 + d_2)$ and is equal to

$$\Lambda = \begin{pmatrix} \Phi_1 & \Psi_1 \\ \Psi_1^T & \Phi_2 \end{pmatrix} \tag{202}$$

Therefore, the surrogate matrix has a factor $d_1^2$ less elements than the full matrix. The main point of this section is that any analytic function of $\Sigma$ has the same form of $\Sigma$, for some other values of the factors, and that the same operation applies to $\Lambda$. In the case of the inverse square root, we thus have

$$\Sigma^{-\frac{1}{2}} = \begin{pmatrix} \Phi_1^{\text{isr}} \otimes I_{d_1} & (\Psi_1^{\text{isr}} \otimes I_{d_1}) K \\ K (\Psi_1^{\text{isr}\,T} \otimes I_{d_1}) & I_{d_1} \otimes \Phi_2^{\text{isr}} \end{pmatrix} \tag{203}$$

$$\Lambda^{-\frac{1}{2}} = \begin{pmatrix} \Phi_1^{\text{isr}} & \Psi_1^{\text{isr}} \\ \Psi_1^{\text{isr}\,T} & \Phi_2^{\text{isr}} \end{pmatrix} \tag{204}$$

We note that the knowledge of the surrogate matrix is sufficient to compute the product between the full matrix and any vector. For example, given the gradient vector

$$\nabla \mathcal{L} = \text{Vec} \left( \frac{\partial \mathcal{L}}{\partial W_1}, \frac{\partial \mathcal{L}}{\partial W_2} \right) \tag{205}$$

Using the properties of Kronecker product, we derive the update for a two-layer Tanh MLP without bias

$$\Sigma^{-\frac{1}{2}} \nabla \mathcal{L} = \text{Vec} \left( \frac{\partial \mathcal{L}}{\partial W_1} \Phi_1^{\text{isr}} + \frac{\partial \mathcal{L}}{\partial W_2^T} \Psi_1^{\text{isr}\,T} , \ \Psi_1^{\text{isr}\,T} \frac{\partial \mathcal{L}}{\partial W_1^T} + \Phi_2^{\text{isr}} \frac{\partial \mathcal{L}}{\partial W_2} \right) \tag{206}$$

Figures 11, 12, 13 compare the full and surrogate matrix for a few example MLPs. They show that the eigenvalues of the full and surrogate matrices are identical, although some of the eigenvalues are repeated multiple times in the full matrix.

## M. Hyperparameter settings

Table 2 shows the hyperparameter values for the experiments illustrated in Figures 5, 6. We note that most optimizers are quite robust with respect to small changes of the hyperparameter values. An exception is KFAC, we found that it was very sensitive to the combination of learning rate $\alpha$ and damping $\lambda$. In Table 2, boldface values are set by grid search, while others are set to standard values. The latter include momentum parameters for Adam ($\beta_1 = 0.9$, $\beta_2 = 0.999$) and KFAC ($\beta = 0.9$), and the exponent of Shampoo, set to $p = \frac{1}{4}$ as common to most applications. For SymO, we set the momentum parameter to $\beta_h = 0.95$, as that value seemed to stabilize training in all our experiments (See Figure 10 in Appendix K).

In the case of SymO, we choose an exponentially decreasing learning rate, $\alpha = \alpha_0 \beta^t$. While this choice is uncommon in neural network optimization, it is motivated by the analytical study of a toy quadratic problem, which is detailed in Appendix B. While neural network optimization is very far from a quadratic problem, it may represent a reasonable approximation near a fixed point, and we found that it worked fine in the non-convex optimization problems illustrated in Figures 5, 6. More work is necessary to find out the optimal settings of SymO.

---

**Algorithm 1** Symmetry-based Optimization update (SymO)

---

**Input:** gradient $\nabla\mathcal{L}$, previous factors $\{\Phi_1, \Phi_2, \Psi_1\}_{t-1}$, momentum parameter $\beta_h$
Compute current factors $\{\Phi_1, \Phi_2, \Psi_1\}_t$ using Equations (192),(193),(194)
Compute surrogate covariance matrix $\Lambda$ by Equation (202)
Divide $\Lambda$ by $(1 - \beta_h^t)$ (bias correction)
Compute inverse square root of surrogate matrix $\Lambda^{-\frac{1}{2}}$ by numerical algebra
Compute factors $\{\Phi_1^{\text{isr}}, \Phi_2^{\text{isr}}, \Psi_1^{\text{isr}}\}$ of $\Lambda^{-\frac{1}{2}}$ by Equation (204)
Compute SymO update by Equation (206)
**Return:** SymO update, current factors $\{\Phi_1, \Phi_2, \Psi_1\}_t$

---

*Table 2.* Hyperparameter values used in experiments, for all optimizers, either in the case of linear activations (Figure 5), or Tanh activations (Figure 6). Values in boldface are set by grid search, others are set to standard values.

| | GD | Adam | KFAC | Shampoo | SymO |
|---|---|---|---|---|---|
| Linear activation | $\boldsymbol{\alpha = 0.8}$ | $\boldsymbol{\alpha = 0.01}$ $\beta_1 = 0.9$ $\beta_2 = 0.999$ | $\boldsymbol{\alpha = 1e-5}$ $\boldsymbol{\lambda = 1.3e-4}$ $\beta = 0.9$ | $\boldsymbol{\alpha = 0.8}$ $\boldsymbol{\epsilon = 1e-10}$ $p = \frac{1}{4}$ | $\boldsymbol{\alpha_0 = 0.05}$ $\boldsymbol{\beta = 0.95}$ $\beta_h = 0.95$ |
| Tanh activation | $\boldsymbol{\alpha = 0.8}$ | $\boldsymbol{\alpha = 0.01}$ $\beta_1 = 0.9$ $\beta_2 = 0.999$ | $\boldsymbol{\alpha = 0.006}$ $\boldsymbol{\lambda = 0.017}$ $\beta = 0.9$ | $\boldsymbol{\alpha = 0.8}$ $\boldsymbol{\epsilon = 1e-8}$ $p = \frac{1}{4}$ | $\boldsymbol{\alpha_0 = 0.09}$ $\boldsymbol{\beta = 0.91}$ $\beta_h = 0.95$ |

