# OpenReview forum: "Global curvature for second-order optimization of neural networks"
_ICML.cc/2025/Conference — ICML 2025 poster_

### Official Review · Reviewer_3sTt · 2025-03-07

**Overall Recommendation:** 3

**Summary:**

The submission studies the structure of the "global curvature" of deep networks. The main result is that global matrix quantities such gradient covariance and Hessian (where "global" means the expected values of those matrices under some distribution on the weights) has a specific matrix structure with much fewer free parameters. As a potential application, the submission proposes an optimizer based on this structure and shows it outperforms SGD/Adam on a two-layer teacher/student problem.


## update after rebuttal

The response has addressed my main issues from the `Claims And Evidence` section below. I was worried that the phrasing in the submission was over-claiming the benefit of "global curvature" and the proposed estimator. The proposed wording and additional evidence for the quality of the claimed estimator would clarify that the main benefit is in the structure of the curvature estimator, which is a more appropriate claim given the supporting experiments. With these changes, I think the paper should be accepted.

**Claims And Evidence:**

The main claim of the submission, that the expected gradient covariance has the claimed structure, is well supported. The visualization provided are convincing up to minor comments (see below).

What's less clear is why this global quantity matters, whether for the gradient covariance or Hessian. The submission seems to take for granted that "global" is better than "local" in its introduction, and the only argument in favor of the global approach is that an algorithm is proposed that uses an estimator inspired by the global curvature structure.

> We demonstrate the effectiveness of our approach by running exact second-order optimization on a two-layer MLP and synthetic data

This appears to assume that the optimizer works better because it estimates the global curvature. I strongly disagree, as the evaluation of the proposed estimator is insufficient and potentially misleading (details in Methods And Evaluation Criteria).

The one potential application I can see for the theory in the global curvature is in enabling efficient full-Gaussian posterior approximation in Bayesian inference at a reduced cost. This is mentioned tangentially in the discussion section, but should be expanded upon to make the link obvious.

---

Please change the following sentence
> We demonstrate the effectiveness of our approach by running exact second-order optimization

The preconditioner does not use any second-order information and is instead an "approximation" of the global covariance matrix of the gradients.

**Essential References Not Discussed:**

Not that I am aware of.

**Experimental Designs Or Analyses:**

No strong issue in the experiment design beyond the evaluation issues listed above.

**Methods And Evaluation Criteria:**

### Evaluation of the estimator

My issue is not that the optimization results are not impressive enough. I am not denying that the proposed preconditioner (using a structured approximation given the current gradient) could be a great optimization method. But whether it does appears disconnected from the global curvature discussion at the start of the paper.

The paper claims to test the estimation error of the proposed estimator,
> Appendix J provides additional details, along with an empirical analysis of the estimation error (see Figure 9)

However, Appendix J and Figure 9 do not test whether the estimator is an appropriate estimator of the global curvature. Instead, they test that it produces consistent results. Those could be close, far, or completely unrelated to the global curvature.

The paper also asserts
> We expect the error in the estimation of hyperparameters to decrease with layer size (d1 in this case).

I did not find evidence or mechanism that would explain this.

The estimator claims to be able to estimate the global covariance from a single sample. This is very suspicious for any claim of globality. Taken to its extreme, the claims in the paper do not seem that far from the claim that "in $d$ dimensions, if $x \sim \mathcal{N}(0, I)$, then $\mathrm{diag}(xx^\top)$ is a good estimator of $I$, especially as $d \to \infty$."

Again; my issue is not with the proposed optimizer or whether it works, but on attributing the reason for its performance to the global curvature. The current paper would be improved if the optimizer was instead introduced along the following lines, to make it clear that the connection is tenuous: "We observe that the global curvature is very structured and sparse. It is possible that this structure captures the most important variations in the covariance/Hessian. Therefore we propose to use this structure as a preconditioner for our optimizer instead of the traditional Diagonal ones. It seems to work well."

I would raise my score if the text was changed to explicitly acknowledge this disconnect, and state that
- the "globality" story is an inspiration, but not a justification, for the good performance of the algorithm
- the good performance of the algorithm, being neither exact nor second-order, does not say much about the validity of the "global" perspective in optimization

### Evaluation of the structure

> We highlight that theory predicts the overall structure, rather than the specific numerical values.

As that is the case, the visualizations could be improved as they emphasize the numerical value. This could be done by fitting the factors of the theoretical structured matrix to the observed data to show agreement on the numerical values.

### Evaluation of optimizers

> For SymO, we choose an exponentially decreasing learning rate

The evaluation of the optimizer claims to use a decreasing step-size
for the proposed optimizer. It is not specified whether the other optimizers (SGD, Adam) use a decreasing learning or other schedule. All algorithms should use the same regime; either a tuned step-size schedule or a constant step-size.

**Other Comments Or Suggestions:**

### Additional details

Some explanation, even if high-level, for the following phenomena would help the reader follow the paper.
- §2.2: Why do ReLU activations satisfy fewer invariances than odd activation functions?
- §3: Why does the addition of an extra hidden layer reduces the inter-layer correlations?

### Word choices

- Please change the following sentence
  > We demonstrate the effectiveness of our approach by running exact second-order optimization

  The preconditioner does not use any second-order information and is instead an "approximation" of the global covariance matrix of the gradients.

- The probability distribution used to sample the relevant quantities in the definition of the "global curvature" is not explicitly given. My understanding is that it is out of a desire to remain general as the theory only requires invariance conditions, but having a running example (for example from variational inference) would help the reader.

- The use of the term "curvature matrix" seems heavily overloaded, as for example in §2.1 it has to encompass the Fisher, the Hessian and the covariance of the gradients.

  > Various studies have employed different formulations for the curvature matrix, including the Fisher information matrix, the Gauss-Newton matrix, the Hessian matrix, and the gradient covariance matrix.

  The relationship between the Hessian and the gradient covariance is unclear, and as a result "curvature matrix" does not seem to imply more than "a matrix that can be used as a preconditioner". If "preconditioner" was the intended meaning, I would encourage its usage instead.

- The submission uses the expression "hyperparameters of the curvature" (and "parameters" in §1). I initially did not understand what it meant, as to me a "hyperparameter" describes a user-specified parameter controlling the behaviour of an algorithm. A more appropriate term could be the "factors", as theorem 1 shows that the expected matrix obeys a specific factorization.

### Typos

In Lemma 2.4, I assume $f$ should be a matrix analytic function of $\Sigma$? $f(\Sigma) = v^\top \Sigma v$ appears analytic but Eq. 15 does not make sense.

**Other Strengths And Weaknesses:**

The study of the impact of the invariance of the network on the gradient and curvature matrices appears novel, and the approximation obtained appear impressive. However, I fail to see why this "global" curvature, that enables such a simplification, helps in the context of optimization which typically relies on local quantities which need not share this structure. This disconnect between the main part of the paper and its tentative application in §3.4 onwards (as detailed above) is my main issue with the current submission.

**Questions For Authors:**

Did I misunderstand the evaluation of the estimator?

**Relation To Broader Scientific Literature:**

The paper could be made stronger by discussing the relationship to the Bayesian variational inference literature in more details. This community is likely to appreciate the structure induced by the invariance, because VI maintains an estimate of the posterior distribution in the form of a  mean and covariance which are used to samples weights at every step to compute updates to the are used to sample weights at every step to compute gradients and update the mean and covariance. The standard approaches use a diagonal approximation to the covariance matrix (eg [Blundell et al., Weight Uncertainty in Neural Networks](https://arxiv.org/pdf/1505.05424)) but more expressive families are sometimes used ([e.g. Lin et al., Fast and Simple Natural-Gradient Variational Inference with Mixture of Exponential-family Approximations](https://arxiv.org/abs/1906.02914)

**Theoretical Claims:**

I have not checked the proofs in details.

---

> ### Author Rebuttal · Authors · 2025-03-31
>
> We thank the reviewer for the detail comments that will improve our paper.
> - *Why global curvature*. We agree on the lack of clarity for why global curvature should be used as a preconditioner. We will revise as follows:
>
> Introduction: “The primary motivation for considering global curvature is its efficient computation, which serves as the main contribution of this work. Future research will explore whether global curvature serves as a reliable approximation of local curvature and whether it offers inherent advantages for enhancing convergence in second-order optimization.”
>
> Section 2.1: “We operate under the assumption that global curvature captures meaningful variations in local curvature for optimization purposes.
>
> Discussion: “This work provides preliminary evidence suggesting that global curvature enhances convergence when applied as a preconditioner. During optimization, the distribution of parameters collapses into a set of local solutions, therefore our method is expected to better approximate the local curvature towards the end of training, similar to other methods (e.g. Natural Gradient, Gauss-Newton). A comprehensive analysis of the errors introduced by our approximation will be a subject of future investigation.”
> - *Bayesian inference*. We agree that our method could be valuable for Gaussian posterior approximation in Bayesian inference. We will revise the Discussion: “Bayesian deep learning relies on approximating the posterior over parameters by a Gaussian distribution. The covariance of this distribution is usually approximated using a diagonal or block-diagonal structure. Our work offers a method for efficiently computing the full covariance, which may lead to more accurate Bayesian posterior estimates.”
> - *Use of the word “second-order”*. While “second-order” traditionally refers to the computation of second-order derivatives, several methods classified as second-order use only first-order derivatives, such as Natural Gradient (e.g. KFAC), Gauss-Newton, and Shampoo. Our approach can be applied to any of those methods. Furthermore, the gradient covariance provides second-order moments of the distribution, therefore we believe that it may be also called “second-order”. However, we are open to reconsidering the terminology if the reviewer strongly feels that it constitutes a misuse.
> - *Estimation error of the global curvature and dependence on layer size*. We will replace Fig 9 with a plot of the correlation of the single-model estimate with the average over a large sample of models (N=10,000), and how that depends on layer size d1. As the reviewer suggested, we do not observe a simple increase of correlation with layer size d1. We observe the following numbers (d0,d1,d2 are the layer sizes):
>
> |Layer sizes|correlation|
> |-|-|
> |(100,10,100)|0.66$\pm$0.05|
> |(100,100,100)|	0.75$\pm$0.03|
> |(100, 1000,100)|0.54$\pm$0.03|
> |(100, 10000, 100)|0.56$\pm$0.03|
>
> - *Theory predicts the overall structure only*. In Fig.2,3,4, we wanted to highlight that, while panels A and B differ significantly, both follow the same structure. Furthermore, the theory of Section 3 remains valid regardless of the specific values of the unknown factors.
> - *Decreasing step size*. We optimize two hyperparameters for all optimization methods except GD (in addition to Adam and SymO, we added Shampoo and KFAC, see answers to other reviewers), ensuring a fair comparison. The choice of an exponentially decreasing learning rate for SymO is based on a theoretical analysis of a quadratic loss function, which we will include in the Appendix.
> - *Why do ReLU activations satisfy fewer invariances?* The intuitive reason is that the symmetry group is smaller, the constraints that are satisfied by the covariance are fewer and therefore the covariance has more degrees of freedom. We will report the groups sizes in Section 2.2.
>  - *Why an extra hidden layer reduces the inter-layer correlations?* Odd activations introduce an invariance with respect to sign changes in both the incoming and outgoing weights of a neuron. We speculate that this invariance may lead to cancellations in the correlations of these weights. Further studies are needed to answer this question.
> - *What is the probability distribution?* We agree that we do not know the distribution of the gradients, other than its symmetries and the structure of its first two moments. We will add an analytic study of a quadratic loss to the Appendix, for which the parameter and gradient distributions remain Gaussian.
> - *Use of the word “curvature”*. Curvature is frequently linked to Fisher Information and Gauss-Newton methods, which do not directly compute the Hessian. The gradient covariance is essentially equivalent to the “empirical” Fisher Information matrix, which is also considered a form of curvature. Nevertheless, we are open to changing the terminology if the reviewer believes it is a misuse.
> - *Use of the word “hyperparameter”*. We agree and we will replace it with “factors”.

---

> > ### Comment · Reviewer_3sTt · 2025-04-07
> >
> > Thanks for the response. I appreciate the check of the large width claim, and the response addresses most of my concern. The proposed modifications would make for a stronger submission. Below are some remaining issues which mostly center around word use.
> >
> > **Why global curvature**
> >
> > These explanations would definitely help, although I am still somewhat uneasy about the phrasing of "the global curvature", since what is being used is the structure of the global curvature, which is imposed on the local curvature to enable efficient estimation.
> > The proposed method does not compute any global quantity, which still seems implied by the above phrasing
> > ("global curvature captures meaningful variations in local curvature", "global curvature enhances convergence when applied as a preconditioner").
> > I would recommend the following edits to emphasize that the proposal is merely to use the factorization derived from the global quantity to approximate a local quantity.
> >
> > > Section 2.1: “We operate under the assumption that global curvature captures meaningful variations in local curvature for optimization purposes.
> >
> > "We operate under the assumption that the structure present in the global curvature is also present in the local curvature and captures meaningful variations for optimization purposes"
> >
> > > “This work provides preliminary evidence suggesting that global curvature enhances convergence when applied as a preconditioner"
> >
> > "This work provides preliminary evidence suggesting that the structure of the global curvature enhances convergence when applied as a preconditioner using local quantities"
> >
> > **Use of the word "second-order"**
> >
> > I grant that some section of the community uses "second-order" to talk about method that do not use second-order information, and even for method which have barely any connection to second-order methods. However, my issue here was more with the wording of "exact second-order", which I would understand to mean the exact computation of the Hessian. Computing the covariance might give some approximation of the second-order information under some specific assumptions, but not the exact one.

---

> > > ### Author Response · Authors · 2025-04-08
> > >
> > > Thank you for taking the time to provide additional comments.
> > >
> > > We believe that we understand your point. You are saying that the theory developed in Section 3 predicts only the structure of the global curvature, and estimating the unknown factors by a single model does not necessarily imply that we compute any global quantity. Therefore, we cannot claim that we compute any global quantity.
> > >
> > > We agree with this point. Even if we have shown that some of the factors estimated by a single model have nearly perfect alignment with the global ones (please see the latest answer to reviewer bCpH), that does not hold for other factors, and more work is needed to understand how much information about global curvature we can obtain with a single model.
> > >
> > > Therefore we will include the changes suggested by the reviewer in the final version of the paper, thank you for pointing those out.
> > >
> > > We also agree with the use of the word "exact second-order". We should not claim that we compute the curvature exactly, because we neither compute the Hessian nor we compute its factors exactly, we only compute its structure. Therefore, for all instances of the word "exact" in the paper, we will either remove it or point out that "exact" refers to the structure only, and we do not compute exactly any second-order information.
> > >
> > > We hope that the reviewer now agrees that our work deserve publication, thank you.

---

### Official Review · Reviewer_ewk1 · 2025-03-09

**Overall Recommendation:** 3

**Summary:**

The work attempts to improve the computations of second order methods by analyzing the covariance matrix of the gradients in small MLP networks. They rely on certain symmetrics expected to be in network parameters and derive theory on the structre, as well as explicit solutions for the covariance matrix. They perform minor experiments, very toyish, where their predictions approximately coincide with the general obtain structure on average (over 10K-100K models). They show on this specific toy example accelaration is obtained, compared to standard GD and Adam optimizers. The experiments are not convincing enough, the setting is highly restrictive, "the input is sampled from a Gaussian distribution with zero mean. The covariance matrix of the input is generated using random orthogonal eigenvectors". It is very hard to conclude if this behavior is actually general and portrays well large NNs and true data. They perform experiments on 3-layer MLP's with RelU, which can already perform quite well on real (easy) data, such as MNIST. Unfortunately, only a toy example is shown. Thus, although the theory is of some interest, I cannot recommend publication in this form of the work.

**Claims And Evidence:**

Claim covariance matrix structure can be predicted, based on certain expected symmetries of the parameters and properties of the input data.

**Essential References Not Discussed:**

refs ok

**Experimental Designs Or Analyses:**

Very scarce, no real analysis. Do not show instances of a certain model, only an average over 10K or 100K models, do not show the variance of their prediction.

**Methods And Evaluation Criteria:**

Evaluation is of low quality, only a very simple toy example assuming Gaussian distribution input.

**Other Comments Or Suggestions:**

see above

**Other Strengths And Weaknesses:**

Strength: theory and general idea may have merrit.

Weaknesses:
* Many assumptions are needed to derive the results.
* Assumptions are not validated on real examples.
* Experiments are not realistic.
* No variance of the results is shown.
* Solution optimization loss (Fig 5), the authors do not show other 2nd order methods, only GD and Adam. This toy examle perhaps works best for 2nd order methods (as it involves Gaussian inputs). Not clear if the proposed solution is faster or simply 2nd order is better here.
* The experiments are perfromed with full gradient of the entire training set, how SGD works? Can the proposed algorithm work well in mini-batches?

**Questions For Authors:**

Please explain in more detail if additional experiments were conducted that I missed, or justify why the chosen setting is reasonable to reflect real world DNNs and data.

**Relation To Broader Scientific Literature:**

little.

**Theoretical Claims:**

Theory on covariance matrix of the gradients. Appear fine.

---

> ### Author Rebuttal · Authors · 2025-03-31
>
> We thank the reviewer for the comments, which will help improving our work.
>
> We would like to highlight that the main contribution of our work is theoretical. As acknowledged by the other reviewers, the theory presented in our work is novel and it could be of broad interests for other fields of machine learning. We hope that we can convince the reviewer of the value and novelty of our theory.
>
> Specific concerns:
> - *Comparison with other second-order methods*. We agree with the reviewer that comparison with other second-order methods is appropriate. We implemented KFAC and Shampoo. In KFAC, we optimised learning rate and damping. In Shampoo, we optimised learning rate and epsilon. The following table shows test loss values for a few time steps in the 2-layer MLP with Tanh case showed in the paper. We will include the full plot in the final version of the paper.
>
> | time      |    GD     |     Adam     |     KFAC    |      Shampoo    |      SymO |
> |----------------|-------------|----------|----------------|-------------|----------|
> | t=50	    |             0.0378 |    0.0255    |     0.0091   |     0.0071       |       0.0058 |
> | t=100 | 0.0232  |  0.0116    |     0.0074     |   0.0060      |        0.0056 |
> |t=150 | 0.0177 |   0.0086    |     0.0067    |    0.0057      |        0.0056  |
> | t=200 | 0.0147 |  0.0072       |  0.0063     |   0.0056       |       0.0056 |
>
> We believe that SymO has better results because it goes beyond block-diagonal preconditioning, and considers also interactions among the two layers. However, the performance of KFAC and Shampoo is very similar to SymO.
>
> - *Relation to broader scientific literature*. The reviewer thinks that our work bears little relation to broader scientific literature, which is the opposite of what stated by the other reviewers, who recognised important connections of our work with other fields of machine learning. We kindly ask the reviewer to specify what references are missing in our paper.
>
> - *How the method works with mini-batches*. We agree with the reviewer that this is an important question. Typically, second-order methods work better with large batches, while first-order methods sometimes work better with small batches. However, the advantage of first-order methods on small batches concerns the generalisation performance of the optimum. Instead, here we are concerned with the speed of convergence, for which no advantage of small batches has ever been observed. Therefore we chose to analyse the full batch case, and postpone the study of mini-batches to future work.
>
> - *Toy examples*. We would like to clarify that the theory provided in Section 3, which is our main contribution, is valid for any size of a neural network. While Figures 2,3,4 report the case of a neural network with 5 neurons per layer, the choice of 5 neurons is only for illustrative purposes and the same theory is valid for any number of neurons.
>
> - *Variance of prediction*. We agree with the reviewer that the we should provide the quality of the estimation of the curvature with a single model. We computed the correlation of the single-model estimate with the estimate obtained by an average over a large sample of models (N=10,000). We observe the following numbers (d0,d1,d2 are the layer sizes):
>
> | Layer sizes	|		correlation |
> |----------------|-------------|
> |(100,10,100)	|		0.66 $\pm$ 0.05 |
> |(100,100,100)	|	0.75 $\pm$ 0.03 |
> |(100, 1000,100)	|	0.54 $\pm$ 0.03 |
> |(100, 10000, 100) |	0.56 $\pm$ 0.03 |
>
> We will include a plot with more details in the final version of the paper.
>
> - *Assumptions are not validated*. We highlight that the theory provided in Section 3, which is our main contribution, is exact and there are no assumptions involved, other than the invariances that are known to hold exactly for neural networks and initialisation routines. We kindly ask to clarify which assumptions the reviewer is concerned about.
>
> - *Experiments are not realistic* We highlight that the main contribution of our work is the theory provided in Section 3. We consider the empirical evaluation as preliminary and not a major contribution of the work. We believe that a full empirical evaluation on a large scale problem deserves a separate paper.
>
> We would like to emphasize the novelty of our theory, as it introduces a unique approach to determining the curvature of neural networks by leveraging their inherent symmetries — a perspective not previously explored. The theoretical methods we developed to tackle this problem are not only original but also hold significant potential for advancing other fields. The study of neural network curvature and loss landscapes is a topic of broad interest, and our work offers new avenues for gaining deeper insights into this major unsolved challenge. We sincerely hope the reviewer will acknowledge the substantial value of our theoretical contribution.

---

> > ### Comment · Reviewer_ewk1 · 2025-04-04
> >
> > Thanks for the detailed rebuttal response and additional experiments. I agree the theoretical part is of significance, assuming the clarifications and experiments additions are incorporated to the paper, I raise my ranking.

---

### Official Review · Reviewer_bCpH · 2025-03-10

**Overall Recommendation:** 4

**Summary:**

The authors' work focuses on getting insights on the second moments of $\Sigma_t  = \int \nabla \mathcal L(\theta)\nabla \mathcal L(\theta)^\top dp_t(\theta) - \mu_t \mu_t^\top$, by exploiting invariances in representation space. In particular, the authors observe that if $G$ is a symmetry of the loss function $\mathcal L(G\theta_t) = \mathcal L(\theta_t)$, then mean and covariance satisfy eigenvalue equations $\mu_t = G \mu_t$, $\Sigma_t = (G^\top \otimes G) \Sigma_t$.
By exploiting symmetry groups $\mathbb G$ of common activation functions, the authors use the fact that these eigenvalue equations have to hold simultaneously for all $G \in \mathbb G$ to show which structure is imposed by the solution space.

**Claims And Evidence:**

I believe the claims of the authors are supported by sufficient empirical evidence.

**Essential References Not Discussed:**

To my knowledge, the idea is novel in this context and the relevant literature has been discussed.

**Experimental Designs Or Analyses:**

In section 3.4, I would include in the comparison also some local preconditioning methods other than Adam, to see in practice if the average structure gives comparable results with respect to a local precondition. In particular, I would compare with [1,2,3]

[1] J. Martens et al., "Optimizing Neural Networks with Kronecker-factored Approximate Curvature", NeurIPS 2023.
[2] N. Vyas et al., "SOAP: Improving and Stabilizing Shampoo using Adam",  ArXiv 2024
[3] V. Gupta et al., "Shampoo: Preconditioned Stochastic Tensor Optimization", ICML 2018.

**Methods And Evaluation Criteria:**

I believe the proposed experimental evaluation suffices for the authors' claims. See "Experimental Designs Or Analyses" for the only concern.

**Other Comments Or Suggestions:**

I believe a point of strong interest would be to quantify how much the second moment in a point $\nabla L(\theta_t)\nabla L(\theta_t)^\top $ can deviate from the average $\mathbb E_{\theta_t}[\nabla L(\theta_t)\nabla L(\theta_t)^\top]$. While not easy to perform since it depends on the update rule, a result of this kind would quantify the local effectiveness of $\Sigma_t$ as a preconditioner for optimization.
Even just experimental investigation in this direction would be of interest.

**Other Strengths And Weaknesses:**

The work is extremely original and the results are of serious interest both from a theoretical point of view and concerning potential applications. In particular, the idea of exploiting neural network's symmetries to infer the structure of the averaged Hessian is of broad interest to develop cheap preconditioners to accelerate training.

**Questions For Authors:**

1. It is interesting to see how your predictions for the structure of the Hessian are extremely sparse in figure 7 (Tanh activation) and less in figure 8 (ReLU activation). Do the authors have any insights about this observation?

**Relation To Broader Scientific Literature:**

The key contribution of this work is theoretical, quantifying a precise relationship between symmetries of the neural network and the averaged Hessian structure. The proposed preconditioning strategy resulting from the theoretical investigation sounds well-developed (no inversions of big matrices), placing also the work in the line of works proposing Quasi-Newton methods for neural network training. As mentioned above, experimental comparison with some of these methods is missing in the manuscript.

**Theoretical Claims:**

I confirm that I checked the proofs in the appendix of the manuscript and I didn't find any criticality.

---

> ### Author Rebuttal · Authors · 2025-03-31
>
> We thank the reviewer for the positive comments. We are glad that the reviewer recognises that the key contribution of our work is theoretical and considers our work novel and interesting.
> We answer here the main concerns raised by the reviewer.
> - *Empirical comparison with other second-order methods*. We implemented KFAC and Shampoo. We used the original version of Shampoo with power 1/4. Using more recent versions with power 1/2 (as in SOAP) did not give better results in our experiments. In KFAC, we optimised learning rate and damping. In Shampoo, we optimised learning rate and epsilon. The following table shows test loss values for a few time steps in the 2-layer MLP with Tanh case showed in the paper. We will include the full plot in the final version of the paper.
>
> | time      |    GD     |     Adam     |     KFAC    |      Shampoo    |      SymO |
> |----------------|-------------|----------|----------------|-------------|----------|
> | t=50	    |             0.0378 |    0.0255    |     0.0091   |     0.0071       |       0.0058 |
> | t=100 | 0.0232  |  0.0116    |     0.0074     |   0.0060      |        0.0056 |
> |t=150 | 0.0177 |   0.0086    |     0.0067    |    0.0057      |        0.0056  |
> | t=200 | 0.0147 |  0.0072       |  0.0063     |   0.0056       |       0.0056 |
>
> We believe that SymO has better results because it goes beyond block-diagonal preconditioning, and considers also interactions among the two layers. However, the performance of KFAC and Shampoo is very similar to SymO.
> - *Sparsity of Hessian for Tanh versus ReLU*. Our understanding of the stronger sparsity in the Tanh case with respect to the ReLU case comes from the symmetry groups. In the case of Tanh, in addition to the permutation symmetry, there is also a symmetry for switching the sign of parameters in adjacent layers. Therefore, the symmetry group of Tanh has larger size with respect to ReLU (please see Chen et al 1993 for specific numbers). As a consequence, the number of constraints that need to be satisfied are larger and the Hessian has less degrees of freedom. These constraints result in a higher sparsity of the matrix. In the final version of the paper, we will report the specific groups  sizesfor Tanh and ReLU in Section 2.2.
> - *Difference between local and global (average) gradient outer product*. It is not easy to compare the second moment at a single point with the ensemble average, because the former is a rank-1 matrix while the ensemble average is usually full rank. Instead, we provide an additional analysis showing the quality of the estimation of the curvature with a single model. We computed the correlation of the single-model estimate with the estimate obtained by an average over a large sample of models (N=10,000). We observe the following numbers (d0,d1,d2 are the layer sizes):
>
> |Layer sizes|correlation|
> |-|-|
> |(100,10,100)|0.66$\pm$0.05|
> |(100,100,100)|	0.75$\pm$0.03|
> |(100, 1000,100)|0.54$\pm$0.03|
> |(100, 10000, 100)|0.56$\pm$0.03|
>
> We will include a plot with more details in the final version of the paper.

---

> > ### Comment · Reviewer_bCpH · 2025-04-02
> >
> > I would like to thank the authors for the thorough response. In order:
> >
> > 1. I appreciate the comparison with other second-order methods and indeed in light of these results (as I was expecting) your theory could be effectively applied to design effective and cheap preconditioners that go beyond diagonal or block diagonal.
> > Most importantly, as far as I know, the idea of using invariances to infer the structure of the Fischer curvature matrix is novel.
> >
> > 2. *Difference between local and global (average) gradient outer product*: I believe it is worth also a discussion about these results since the behavior of correlation seems to be a bit "erratic" on $d_1$ and I cannot find a reason for this (aligned with reviewer 3sTt comment). I would appreciate it if the authors could expand on this point, at least giving some insights.
> >
> > In any case, I am very satisfied with the rebuttal and therefore I will maintain my score.

---

> > > ### Author Response · Authors · 2025-04-02
> > >
> > > Thank you, indeed we also did not understand the dependence of correlations on d1, we thought that correlations should just increase with d1. After a more careful analysis, we now understand what is happening.
> > >
> > > We broke down correlations in different blocks of the curvature matrix, and we made two crucial observations that explain the numbers provided in our previous rebuttal: 1) Correlation is different in different blocks, and increases with d1 for all blocks 2) Frobenius norm is different in different blocks, and decreases with d1 for all blocks. The increase in correlation and decrease of Frobenius norm explains a non-monotonic correlation when considering the full curvature matrix with all blocks.
> > >
> > > We consider three blocks: B11 (layer-1 to layer 1), B12 (layer-1 to layer 2), B22 (layer 2 to layer 2).
> > > The correlations monotonically increase with d1 for all blocks
> > >
> > > |Layer sizes           |B11                  |B12                  |B22                  |
> > > |-|-|-|-|
> > > |(100,10,100)         |0.67$\pm$0.05|0.38$\pm$0.06|0.30$\pm$0.05|
> > > |(100,100,100)       |0.90$\pm$0.02|0.61$\pm$0.04|0.52$\pm$0.03|
> > > |(100, 1000,100)    |0.96$\pm$0.01|0.64$\pm$0.03|0.54$\pm$0.03|
> > > |(100, 10000, 100) |0.97$\pm$0.01|0.65$\pm$0.03|0.56$\pm$0.03|
> > >
> > > Furthermore, correlations in B11 are higher while correlations for B22 are lower.
> > > However, the Frobenius norm decreases with d1, and has markedly different values in different blocks
> > >
> > > |Layer sizes           |B11       |B12        |B22        |
> > > |-|-|-|-|
> > > |(100,10,100)         |1.0684   |0.1119   |0.0203   |
> > > |(100,100,100)       |0.0321   |0.0113   |0.0149   |
> > > |(100, 1000,100)    |0.0024   |0.0011   |0.0129   |
> > > |(100, 10000, 100) |0.0002   |0.0001   |0.0131   |
> > >
> > > For small values of d1, the norm of B11 dominates, thus correlations in the full matrix are high. For larger values of d1,  the norm of B22 dominates, thus correlations in the full matrix are low. We hope that this new analysis clarifies the issue.

---

### Decision · Program_Chairs · 2025-05-01

**Decision:**

Accept (poster)

**Comment:**

This work introduces a theoretical framework that leverages neural network symmetries for efficient and exact computation of the covariance matrix of gradients, which can be used in scaled-gradient optimization methods. By exploiting these symmetries, the computation of such matrices involves fewer degrees of freedom than naively expected. The approach reveals structural insights into curvature, influenced by activation functions, and demonstrates faster convergence on synthetic data.

The paper addresses an interesting problem and offers novel perspectives and solutions. Reviewers have provided several suggestions regarding terminology (specifically the use of "second-order"), the scope and depth of the claims, the underlying assumptions, and the experimental section. The authors have addressed many of these in their rebuttal and are encouraged to incorporate them in the next revision of the paper.